# Epidemiology of malaria, schistosomiasis, and geohelminthiasis amongst children 3–15 years of age during the dry season in Northern Cameroon

Francis N. Nkemngo[1,2]*, Lymen W. G. Raissa[1], Derrick N. Nebangwa[3], Asongha M. Nkeng[4,5], Alvine Kengne[6], Leon M. J. Mugenzi[1], Yvan G. Fotso-Toguem[1], Murielle J. Wondji[1,7], Robert A. Shey[8], Daniel Nguiffo-Nguete[1], Jerome Fru-Cho[2,11], Cyrille Ndo[1,9], Flobert Njiokou[6], Joanne P. Webster[10], Samuel Wanji[2,11], Charles S. Wondji[1,7]*

1 Centre for Research in Infectious Diseases (CRID), Yaoundé, Cameroon, 2 Department of Microbiology and Parasitology, Faculty of Science, University of Buea, Buea, Cameroon, 3 Faculty of Life Science and Medicine, King's College London, New Hunt's House, London, United Kingdom, 4 Centre for Infection Biology and Translational Research (CIBiT), Forzi Institute, Buea, Cameroon, 5 Department of Sociology & Anthropology, Faculty of Social and Management Sciences, University of Buea, Buea, Cameroon, 6 Department of Animal Biology and Physiology, Parasitology and Ecology Laboratory, Faculty of Science, University of Yaoundé I, Yaoundé, Cameroon, 7 Vector Biology Department, Liverpool School of Tropical Medicine, Pembroke Place, Liverpool, United Kingdom, 8 Department of Biochemistry and Molecular Biology, Faculty of Science, University of Buea, Buea, Cameroon, 9 Department of Biological Sciences, Faculty of Medicine and Pharmaceutical Sciences, University of Douala, Douala, Cameroon, 10 Department of Pathobiology and Population Sciences, Royal Veterinary College, University of London, Herts, United Kingdom, 11 Research Foundation in Tropical Diseases and Environment (REFOTDE), Buea, Cameroon

* francis.nkemngo@crid-cam.net, nkemngo.francis@gmail.com (FNN); charles.wondji@lstmed.ac.uk (CSW)

**Data Availability Statement:** The minimal dataset for this submission has been uploaded in the OSF

## Abstract

### Background

The double burden of malaria and helminthiasis in children poses an obvious public health challenge, particularly in terms of anemia morbidity. While both diseases frequently geographically overlap, most studies focus on mono-infection and general prevalence surveys without molecular analysis. The current study investigated the epidemiological determinants of malaria, schistosomiasis, and geohelminthiasis transmission among children in the North Region of Cameroon.

### Methodology

School and pre-school children aged 3–15 year-of-age were enrolled from three communities in March 2021 using a community cross-sectional design. Capillary-blood samples were obtained, and each was examined for malaria parasites using rapid-diagnostic-test (RDT), microscopy, and PCR while hemoglobin level was measured using a hemoglobinometer. Stool samples were analyzed for *Schistosoma mansoni*, *S. guineensis*, and soil-transmitted-helminthiasis (STH) infections using the Kato Katz method, and urine samples were assessed for the presence of *S. haematobium* eggs (including hybrids) using the standard urine filtration technique.

repository: https://osf.io/cbdm6 and Identifier: DOI
10.17605/OSF.IO/FH6MR.

**Funding:** FNN received a Joint Royal Society of
Tropical Medicine & Hygiene (RSTMH) and
National Institute of Health Research (NIHR) early
career research grant, https://rstmh.org/grants/
grant-awardees-2020/nihr-awardees-2020. The
funders had no role in study design, data
collection, and analysis, decision to publish, or
preparation of the manuscript.

**Competing interests:** he authors have declared
that no competing interests exist.

## Result

A malaria prevalence of 56% (277/495) was recorded by PCR as opposed to 31.5% (156/495) by microscopy and 37.8% (186/495) by RDT. Similarly, schistosomiasis was observed at prevalence levels of up to 13.3% (66/495) overall [*S. haematobium* (8.7%); *S. mansoni* (3.8%); mixed *Sh/Sm* (0.6%); mixed *Sh/Sm/Sg* (0.2%). Both infections were higher in males and the 3–9 year-of-age groups. A high frequency of PCR reported *P. falciparum* mono-infection of 81.9% (227/277) and mixed *P. falciparum/P. malariae* infection of 17.3% (48/277) was observed. Malaria-helminths co-infections were observed at 13.1% (65/495) with marked variation between *P. falciparum/S. haematobium* (50.8%, 33/65); *P. falciparum/S. mansoni* (16.9%, 11/65) and *P. falciparum/Ascaris* (9.2%, 6/65) ($\chi^2$ = 17.5, p = 0.00003). Anemia prevalence was 32.9% (163/495), categorically associated with *P. falciparum* (45.8%, 104/227), Pf/Sh (11.5%, 26/227), and Pf/Sm (3.9%, 9/227) polyparasitism.

## Conclusion

Polyparasitism with malaria and helminth infections is common in school-aged children despite periodic long-lasting insecticide-treated nets (LLINs) distribution and regular school-based praziquantel (for schistosomiasis) and albendazole (for STH) campaigns. Co-existence of *Plasmodium* parasites and helminths infections notably *Schistosoma* species among children may concurrently lead to an increase in *Plasmodium* infection with an enhanced risk of anemia, highlighting the necessity of an integrated approach for disease control interventions.

## Introduction

Despite the implementation of control measures since the early 2000s, malaria and helminth infections rank as the most prevalent parasitic diseases in Cameroon where they impose a tremendous public health burden, particularly on children [1, 2]. Malaria was responsible for 6,840,000 cases and more than 3,500 deaths in 2019, with *P. falciparum* accounting for approximately 95% of the attributed deaths in the country [2, 3]. Additionally, the circulation and expansion of non-*P. falciparum* species, particularly *P. malariae* and *P. ovale* [4], often co-existing with *P. falciparum* have been reported to widen the force of transmission and increase the magnitude of disease severity with anemia being the most profound outcome [5]. The disease exerts a significant impact on morbidity and mortality, particularly amongst infants and children under five (0–5 years) due, at least in part, to minimal acquired protective immunity [3]. As such, interventions such as long-lasting insecticide-treated nets (LLINs), artemisinin combination therapy (ACTs), intermittent preventive treatment for infants (IPTi), seasonal malaria chemoprevention (SMC), and potential vaccination programs are concentrated on this under five years-of-age group to significantly reduce the risk of death [3]. However, whilst malaria-associated mortality is relatively lower amongst school-aged children (SAC) (5–15 years), this age group also represents a key epidemiological demographic grouping [6]. Due to repeated exposure to malaria and acquisition of partial immunity to disease in high transmission settings, SAC are frequently subjected to the prolonged carriage of untreated asymptomatic parasitemia, thereby serving as potential infectious reservoirs for sustaining transmission [7]. From a logistical perspective, primary school children also tend to be easily accessible for

large scale surveillance of a range of infectious diseases including malaria. Moreover, with the changing pattern and dynamics of malaria transmission in Cameroon following the scale-up of national control interventions since 2015, monitoring the infection profiles and morbidity in the SAC age group remains fundamental to evaluating the success of malarial control programs.

Furthermore, evidence from epidemiological studies to date has demonstrated that in areas of co-endemicity, children exposed to malaria often have a high likelihood of helminth co-infections [5]. Helminthic infections particularly schistosomiasis (*Schistosoma haematobium* (and hybrids therein), *S. mansoni*, and *S. guineensis*) and soil-transmitted worms (*Ascaris lumbricoides*, *Ancylostoma duodenale*, *Necator americanus*, and *Trichuris trichiura*) are chronic debilitating neglected tropical diseases prevalent in Cameroon, and affecting more than 200,000 people [8, 9]. Like malaria, these diseases thrive mostly in poor and rural communities where the vulnerable groups suffer from anemia, reduced productivity, and poor psychological development [10]. Transmission of the diseases is primarily governed by social-ecological factors where access to safe water, poor latrine systems, inadequate hygiene, and sanitation practices combined with domestic activities predispose the high-risk groups, particularly children to water contacts or infected soils harboring the infective larva forms of the parasites [11–13]. Pathology of these diseases notably schistosomiasis is largely mediated by the eggs which are often excreted in urine or feces to ensure the continuity of the parasite life cycle [14]. Preventive chemotherapy (PCT) through school-based periodic administration of praziquantel (PZQ) and albendazole (ABZ) to school-age children remains the cornerstone of schistosomiasis and STH control respectively in endemic foci in Cameroon [8, 9]. While this strategy reduces the egg load and minimizes transmission, it does not prevent infection or re-infection, thereby posing concerns about the potential long-term interruption of disease transmission [15]. Moreover, the exclusion of infected preschool children and adults from treatment campaigns implies that these groups continually serve as transmission hotspots and may suffer from anemia, organ dysfunction, and serious reproductive and mental health outcomes [10].

In Cameroon, helminth infections, particularly schistosomiasis, are endemic in the Northern Region [8]. However, whilst there is now a substantial body of evidence concerning the prevalence and seasonal transmission of malaria and helminthiasis during the rainy season in the region, there remains limited available data during the dry season. Furthermore, where data exist, the majority of research efforts still focus on the epidemiology of single infections [16], without assessing the epidemiological and morbidity impact of polyparasitism. Thus, here we sought to provide an update on the prevalence of malaria and helminths infections and to comprehensively assess the risk factors associated with the disease transmission in the North Region of Cameroon, with a specific focus on the dry season. These findings will contribute to the design or improvement of policy decision strategies to concomitantly interrupt malaria and helminths transmission in Cameroon.

## Materials and methods

### Ethics approval and consent to participate

The study received ethical approvals (2021/1250-10/UB/SG/IRB/FHS) and (N0 2020/05/1234/CE/CNERSH/SP) from the Ethics Review Board of the Faculty of Health Sciences, University of Buea and the Cameroon National Committee on Ethics for Human Health Research respectively. Administrative authorization was sought from the North Regional Delegations of Public Health and Basic Education, Garoua, Cameroon. The study was conducted conforming to the World Medical Association (WMA) guidelines as highlighted in the Declaration of Helsinki. Sensitization of the population was done in the various communities before the study. The purpose, risks, and benefits of the study were explained to the parents and guardians of

children both in French and the local dialect (Fulfulde), and written informed consent was obtained from all the parents/guardians whose children were enrolled in the study. By signing the form, the children ≤ 5 years-of-age (assisted by their respective guardian/parent) and children > 5 years-of-age agreed to answer a questionnaire and to provide finger-prick blood, urine, and stool samples for parasitological analysis. Confidentiality was respected as participants responded to the questionnaire and even during data processing and analyses, as well as during data sharing. Participation was voluntary, and individuals could freely terminate involvement in the study at any time, even without prior notice. Children who were positive for malaria were administered first-line treatment as recommended by the national treatment guideline policy. However, children positive for schistosomiasis and worm infections were referred to the local district hospital where free praziquantel (40 mg/kg body weight) and albendazole (400 mg) treatments were respectively given, in association with the Cameroon Schistosomiasis Control Program (PNLSHI) [8].

## Study area

This study was conducted in March 2021 during the dry season in three community health districts (HD) of the sudano-guinean climatic zone of the North Region of Cameroon (Fig 1), an area with an average annual precipitation level of <100mm [17]. These communities, mainly dominated by Muslims, include Pitoa Centre (9˚38'95" N, 13˚50'15" E) and Wouro-kessoum (PWK) (9˚38'93" N, 13˚50'14" E) in the Pitoa health district (PHD), Bainga Assoura (9˚85'35" N, 13˚95'68" E) and Kola (BAK) (9˚85'86" N, 13˚95'94" E) harboring a major touristic feature —Georges de Kola- in the Guider health district (GHD) and Gounougou (GNG) (9˚03′00″N, 13˚43′59″E) in the Lagdo health district (LHD). Daily temperature ranges from 28˚C to 35˚C. The choice of selected sites was guided by the geo-cardinal positions (Fig 1) combined with the existence of previously available malaria and helminths parasitological data during the rainy season [18, 19]. The climate of these localities is characterized by a short rainy season from May

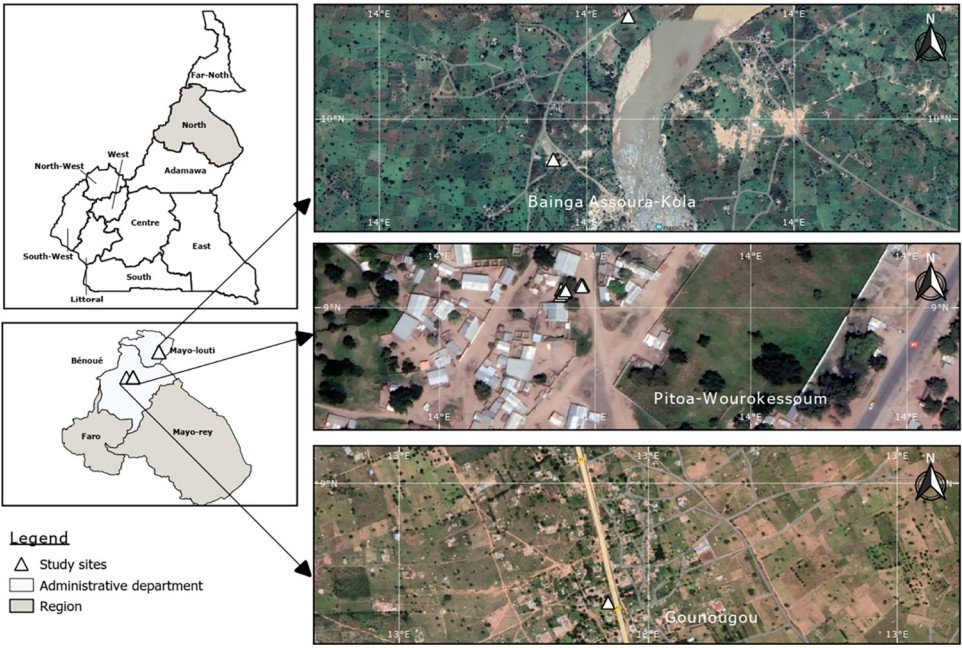

**Fig 1. Study sites in North Cameroon.**

to September (mean annual rainfall of 900 to 1000 mm) and a long dry season from October to April. Farming (cotton and rice cultivation), cattle rearing and fishing are the primary occupation of the inhabitants since these communities are proximal to natural and artificial water bodies including major rivers (e.g., River Benoué), streams, and a hydroelectric power plant (Lagdo dam) [20, 21]. These water bodies constitute the principal breeding sites of the snail intermediate host. Poor infrastructure planning with most houses lacking toilets or water systems and the common practice of open defecation is the norm in these villages. Moreover, the influx of displaced persons from the Boko-Harram conflict-hit areas in the Far-North Region [22] coupled with conditions of poor personal and environmental sanitation facilities, exacerbates the transmission of diseases like schistosomiasis. Indeed, the general lack of latrines and inadequate potable water supply at the community level compels the population to utilize these water bodies for personal, household, and livelihood activities; further risking exposure to *Schistosoma* cercariae. However, sustained annual preventive chemotherapy-based mass drug administration (MDA) of PZQ and ABZ to SAC has been ongoing for over 15 years [8]. Furthermore, malaria transmission is seasonal [2] with the peak period ranging from September to October. *Anopheles coluzzii* and *An. arabiensis* are the main vectors involved with the majority of the households administered an LLIN by the Cameroon National Malaria Control Program at a coverage of ~ 76% during the 2018 mass distribution campaign [2].

## Study design and sample size determination

A community-based cross-sectional design was utilized to recruit—children (3–15 years-of-age) residing in the North Region of Cameroon. The sample size of the study population was calculated based on a previous schistosomiasis prevalence of 38.5% during the rainy season [8] considering that malaria is endemic in this Region. The sample size, N was determined using the Cochran's formulaby: $N = Z^2 \times P (1-P)/d^2$ [23] where Z is the standard normal deviation, Z = 1.96 for the confidence level of 95%, P = 0.385; the proportion of schistosomiasis prevalence, d is the error width of the confidence interval (0.05). The minimum estimated sample size required was calculated as 370. A total of 555 participants were, however, considered for recruitment to anticipate attrition factors such as voluntary withdrawal, and/or to accommodate for the loss of sample due to the inability of all the children to provide capillary blood or urine/stool samples at the time of specimen collection. Each participant provided three specimens including blood spots, stool, and urine samples. Parents and their respective children were invited by local community leaders to assemble at a major focal point in each of the selected communities for sample collection. Health education talks on preventive measures for these diseases were delivered before each screening exercise. Furthermore, a questionnaire was administered to each participant by a trained research assistant to obtain anthropometric and socio-demographic data. As inclusion criteria, only children who had lived for at least a year in the selected villages and whose parents/guardians consented to participate were enrolled in the study. Parents who brought more than one child were registered under the same household (or a different household if the child was not theirs). Those who failed to provide feces or urine samples after the interview were excluded.

## Administration of questionnaire

A simple structured questionnaire (S1 File) was used to collect information about each individual based on the following variables: socio-demographic data (including name, address, age, gender, and educational level), clinical data, malaria risk factors (including knowledge on malaria, history of fever, long-lasting insecticide-treated nets (LLINs) ownership and use, anti-malarial intake), schistosomiasis risk factors (including knowledge on schistosomiasis, source

of water supply, frequency of water contact, open-air defecation practice, hand washing and hygienic practice, deworming history). Overall, a total of thirty-seven variables were assessed with scores either between Yes or No and multiple provided answers (S1 File). These scores were computed to give the total counts of the response to each variable. For example, the KAP scoring system for variables with Yes/No response employed a cut-off binary code digit where 0 indicates poor KAP and 1 denotes good KAP. The basis of the decided cut-off was to categorize the data set and establish relevant epidemiological associations between the level of risk/exposure and outcome.

## Clinical assessment

A digital thermometer was used to measure the axillary temperature and fever was defined as body temperature ≥ 37.5˚C. Participants' heights were measured using a stadiometer (Seca 206IN, Hamburg, Germany) while a weighing balance (Michiels, Luitberg, Belgium) of 120 Kg was used to measure the body weight.

## Parasitological examination of blood samples

Finger prick (capillary) blood sample was collected from each individual to measure hemoglobin (Hb) level using a hemoglobinometer (URIT Medical Electronic Co., Shenzhen, China) and perform malaria rapid diagnostic test (RDT) using the CareStart[TM] Malaria Pf/Pan (HRP2/pLDH) antigen Combo kit following manufacturer's guidelines. Blood aliquots were used to prepare thick and thin films on labeled slides to determine the malaria parasite density using the Giemsa staining technique. Slides were observed under X100 objective lens (oil immersion) of a light microscope [24]. Each blood film was independently examined under the light microscope (limit of detection: 50–100 parasites/μL) by two trained and experienced microscopists, based on established protocols for the detection and identification of *Plasmodium* parasites. Slides were considered positive when asexual stages (trophozoites and schizonts) or gametocytes were identified. Parasite density was determined on thick blood films by systematic counting of parasite numbers against at least 200 white blood cells (WBCs), assuming a standard WBC count of 8000/μL. If gametocytes were detected, the count was extended to 500 leukocytes [24]. Malaria parasitemia was categorized as low (≤ 1,000 parasite/μL of blood), moderate (1,000–5,000 parasites/μL of blood), and high (> 5,000 parasites/μL of blood). Slides were considered negative after scanning through the entire stained smear [24].

Following the same slide and RDT numerical codes, three drops of capillary blood aliquoted onto an ID labeled Whatman 3 mm filter paper were air-dried, kept in a sealed plastic bag, and stored at room temperature for subsequent laboratory analysis. The dried blood spots (DBS) samples were used to detect and identify the malaria parasite species and to determine sub-microscopic *Plasmodium* infection. Genomic DNA was extracted from the DBS using chelex beads [25] while primary and nested polymerase chain reaction (PCR) assays were run to discriminate the *Plasmodium* species. The primary PCR was done using a pair of *Plasmodium* genus-specific primers which amplifies a 1100-base pair (bp) PCR product from the 18S rRNA small subunit gene. Briefly, amplification was performed using rPLU5 and rPLU6 primers to identify the *Plasmodium* genus. The PCR mixture of 20 μl total volume, contained 10X buffer, 200 Mm dNTPs, 10 μM of each primer, 2.5 mM MgCl2, 1 unit of Taq DNA polymerase (Invitrogen, Carlsbad, CA), and 5 μl of extracted DNA. The thermocycling consisted of an initial denaturation step at 95˚C for 5 min, followed by 25 cycles at 94˚C for 30s, 58˚C for 2 min, and 72˚C for 2 min, and a final extension step at 72˚C for 10 min. Similarly, the nested PCR (nPCR) with primers specific to the four human *Plasmodium* species (*P. falciparum*, *P. malariae*, *P. ovale*, and *P. vivax*) were used to amplify each species. The master mix composition was

similar except that 4 μl aliquot of the primary PCR product was used as a template in a cocktail mix of the species-specific primers. The PCR thermocycling conditions were initial denaturation at 95°C for 5 min, followed by 30 cycles each of 30 sec at 94°C, 2 min at 58°C, 2 min at 72°C, and 5 min final elongation at 72°C.

The PF3D7 laboratory strain and distilled water were used as positive and negative controls respectively [26].

## Parasitological examination of urine samples

On the day of enrolment, all parents/guardians of children who participated in the questionnaire sampling received a 50 mL sterile, wide-mouthed; screw-capped plastic containers carrying their identification information for urine collection. Considering the circadian pattern of schistosome egg excretion, participants were requested to collect urine samples between 10 am– 2 pm [27]. Upon receipt of urine, hematuria was immediately determined by visual inspection and urine reagent strips (Uric11V, ACCU-ANSWER®) following the manufacturer's instructions. All urine and stool samples were stored in cooling boxes containing air cooler ice packs at a temperature of about 4°C to prevent *S. haematobium* eggs from hatching and hookworm eggs from degrading during transportation to the laboratory for processing within 12–18 hours of collection. The urine samples were processed using the Nucleopore syringe urine filtration technique and examined microscopically for the presence of *S. haematobium* infection based on the morphology of the ova [28]. Briefly, 10-mL homogenized urine collected using a syringe was forced through a polycarbonate membrane filter (STERLITECH Corporation, USA). The filter was removed from the filter holder and placed on a glass slide using blunt-ended forceps. The slide template was stained with 1% Lugol's Iodine Solution, covered with a cover slip, and examined microscopically under X10 objective of the light microscope. Terminal-spined eggs characteristic of *S. haematobium* were identified and counted manually. The infection intensity was defined by the number of eggs per 10 mL of urine and categorized as light (<50 eggs/10 mL of urine) or heavy (≥50 eggs/10ml of urine) infection as defined by the WHO [29].

## Parasitological examination of stool samples

Similarly, an ID-labeled sterile 50 mL plastic containing the fecal specimen of each enrolled child was submitted for examination. The fresh stool samples were analyzed using the Kato Katz technique employing a 41.7 mg template. Briefly, the feces were pressed through a mesh screen to remove large particles. A portion of the sieved sample was then transferred into the template orifice on each slide. After filling the hole, the template was removed and the sample (~ 41.7 mg) was firmly covered with a piece of cellophane soaked overnight in glycerol-malachite green solution. The glycerol clears the fecal debris from around the eggs. The prepared slides were mounted under a Leica® light microscope and observed under X40 objective to identify the presence of *S. mansoni*, *S. guineensis*, and STH eggs based on identification charts. The fecal smears were examined within 1 hour of preparation to avoid missing hookworm ova. Duplicate smears were prepared for each sample and examined by trained and experienced microscopists. The intensity of infection was expressed as eggs/gram of feces (epg) [30].

## Data analysis

Data collected were entered in MS Excel, imported into and analyzed using GraphPad Prism V8 (GraphPad Software, La Jolla, California USA). Children were categorized into 3–9 and 10–15 years age groups. Descriptive statistics involving continuous variables were summarized into means and standard deviations (SD), and categorical variables were reported as

frequencies. General and species prevalence of malaria and schistosomiasis were calculated as the proportion of individuals that were identified as positive for the presence of parasites (or species) considering the villages, age group, and gender category. Fisher exact test was employed to verify whether a significant association exists between two categorical variables. Also, independent variables were subjected to chi-square ($\chi^2$) analysis for test of association. Variables with a p-value $< 0.25$ underwent step-wise logistic regression analysis for adjusted odds ratios (aOR). Significant levels were measured at a 95% confidence interval (CI) with statistically acceptable differences set at p $< 0.05$.

## Definitions of endpoints

Sub-microscopic infection was defined as low-density blood-stage parasitemia that was not detected by gold-standard microscopy but positive by nested PCR (nPCR). Asymptomatic malaria parasitemia was defined as the presence of *Plasmodium* parasite by microscopy or nPCR with an axillary temperature of $< 37.5°C$ and the absence of fever within the past 14 days while clinical malaria parasitemia was considered as the presence of *Plasmodium*, with an axillary temperature of $\geq 37.5°C$, joint pains, vomiting, headache, diarrhea, chills [31]. Anemia was defined as Hb $< 11.0$ g/dL and further categorized as severe (Hb $< 7.0$ g/dL), moderate (Hb level: 7.0–10.0 g/dL), and mild (10.1 and $< 11$ g/dL) [32]. Symptomatic urogenital schistosomiasis was defined as the presence of *S. haematobium* eggs in the urine associated with dysuria, hematuria, and abdominal pain while non-symptomatic urogenital schistosomiasis as the presence of *S. haematobium* eggs in urine without any visible signs and symptoms of fever, hematuria, abdominal pain, and difficult/painful urination [33]. Similarly, symptomatic intestinal schistosomiasis was defined as the presence of *S. mansoni* eggs in feces with abdominal pain/blood spots in feces as opposed to ectopic egg excretion which is considered as the presence of *S. mansoni* eggs in urine or *S. haematobium* eggs in feces [21].

True positive (TP)–individuals have the disease and test positive; False positive (FP)–individuals do not have the disease but test positive; True negative (TN)–individuals do not have the disease and test negative; False negative (FN)—individuals have the disease but test negative. Sensitivity (Se) of the test is the ability of the test to identify correctly those who have the disease (true positive rate), usually denoted by Se = TP/ (TP + FN) while Specificity (Sp) of the test is the ability of the test to identify correctly those who do not have the disease (true negative rate), designated by Sp = TN/ (TN + FP). Positive predictive value (PPV) is the probability that a disease is present when the test is positive [PPV = TP/ (TP + FP)]. Negative predictive value (NPV) is the probability that the disease is not present when the test is negative [NPV = TN/ (TN + FN)]. Accuracy (Acc) is the overall probability that a patient will be correctly classified [Acc = (TP + TN)/ (TP + TN + FP + FN)] [34].

## Results

### Baseline characteristics of the study participants

A total number of 495 children (3–15 years) who completed questionnaires were enrolled and provided capillary blood, urine, and stool samples for this study. The baseline characteristics of the study population are shown in Table 1. Stratification of the age group showed that 62.6% (310) of the participants were between 3 to 9 years old while 37.4% (185) were within the 10–15 years range. Among the 495 participants enrolled in this study, 45.5% (225) were females while 54.5% (270) were males. The respective mean age and weight for females of the 3–9 years age group were 6 years (±1.8) and 23.8 kg alongside 11.4 years (±1.4) and 37.7 kg for the 10–15 years age group (Table 1). Similarly, the mean age and weight for males of the 3–9 years group was 5.9 years (±1.8), 24.9 kg with the 10–15 group recording 11.7 years (±1.5), 37.4 kg Females

**Table 1. Characteristics of study participants by sex and age in the three communities.**

| Category | | Age (yr) | | Total |
|---|---|---|---|---|
| Variables | | 3–9 | 10–15 | |
| % (n) | | 62.6 (310) | 37.4 (185) | 100 (495) |
| Sex | Female | 64.4 (145) | 35.6 (80) | 45.5 (225) |
| | Male | 61.1 (165) | 38.9 (105) | 54.5 (270) |
| Locality | Pitoa & Wourokessoum | 58.1 (132) | 41.9 (95) | 45.9 (227) |
| | Bainga Assoura & Kola | 67.8 (80) | 32.2 (38) | 23.8 (118) |
| | Gounougou | 65.3 (98) | 34.7 (52) | 30.3 (150) |
| Mean age (yr ± SD) | | 5.9 ±1.8 | 11.4±1.4 | 8.7±1.6 |
| Mean height (m ± SD) | | 1.2±0.1 | 1.4±0.1 | 1.3±0.1 |
| Mean weight (kg ± SD) | | 23.8±6.6 | 37.7±8.1 | 30.7±7.3 |
| Mean body temp (˚C) | | 36.8 | 36.8 | 36.8 |
| Mean Hb (g/dL) | | 11.1 | 12.0 | 11.6 |
| *School enrollment | Yes | 51.5 (189) | 48.5 (178) | 86.8 (367) |
| | No | 87.5 (49) | 12.5 (07) | 13.2 (56) |
| Fever | Yes | 75.0 (27) | 25.0 (09) | 7.3 (36) |
| | No | 61.7 (283) | 38.3 (176) | 92.7 (459) |
| History of fever in the past 3 days | Yes | 33.3 (03) | 66.7 (06) | 1.8 (09) |
| | No | 63.2 (307) | 36.8 (179) | 98.2 (486) |

*School enrollment excluded unregistered preschoolers (children ≤ 5 years); n = 423

(87.6%) with body temperature $< 37.5$˚C were more than males ($\chi^2 = 17.79$, p = 0.02). At the time of the survey, 13.2% of children were currently not enrolled in the local basic educational system, accounting for the literacy gap observed in this region.

## Prevalence of malaria and associated risk factors among study participants

Out of the 495 participants examined by microscopy, 31.5% (156/495) tested positive for malaria parasites (Table 2). Generally, males (34.8%, 94/270) recorded a higher parasite prevalence than females (27.6%, 62/225) ($\chi^2 = 2.99$, p = 0.08). Considering the sample size, children of 3–9 years were similarly infected (30.6%, 95/310) as the 10–15 years (32.9%, 61/185) age group although no significant difference in malaria prevalence was observed ($\chi^2 = 0.29$, p = 0.59). Among females, the 3–9 years age group had the highest malaria prevalence of 28.9% (42/145). Likewise, for the male gender, 32.1% (53/165) prevalence was observed in the 3–9 years category and 39% (41/105) in the 10–15 age cohort with no significant difference observed ($\chi^2 = 1.36$, p = 0.24). *P. falciparum* was the most dominant malaria causative species accounting for 62.1% (59/95) and 72.1% (44/61) of cases in the 3–9 and 10–15 age groups respectively. Mix infection of *P. falciparum* and *P. malariae* (*Pf+Pm*) was pronounced in the 3–9 years age group (10.5%, 10/95) and male gender (6.4%, 6/94) ($\chi^2 = 13.3$, p = 0.04). Mean *P. falciparum* trophozoite density was higher in males [1351 parasites/μL; (440–13040)] and the 3–9 years age group [1426 parasites/μL; (440–49840)]. Contrary to microscopy, RDT and nPCR reported a malaria prevalence of 37.6% (186) ($\chi^2 = 4.02$, p = 0.04) and 55.9% (277) ($\chi^2 = 60.1$, p < 0.0001) respectively as shown in Table 2. The proportion of the infected population by gender varied between female [36.9% (83) RDT vs 58.2% (131) nPCR] ($\chi^2 = 46.2$, p < 0.0001) and male [38.1% (103) RDT vs 54.1% (146) nPCR] ($\chi^2 = 29.4$, p < 0.0001). Similarly, the 3–9 years age group documented higher prevalence [36.8% (114) RDT vs 56.1% (174) nPCR] ($\chi^2 = 43.4$, p < 0.0001) than the 10–15 years [38.9% (72) RDT vs 55.7% (103) nPCR]

**Table 2. Malaria prevalence, parasitemia indices and associated risk factors by study communities.**

| Category | | Age (yr) | | Total |
|---|---|---|---|---|
| **Variables** | | 3–9 | 10–15 | |
| **% (n)** | | 62.6 (310) | 37.4 (185) | 100 (495) |
| **Malaria prevalence** | RDT | 61.3 (114) | 38.7 (72) | 37.6 (186) |
| | Microscopy | 60.9 (95) | 39.1 (61) | 31.5 (156) |
| | PCR | 62.8 (174) | 37.2 (103) | 55.9 (277) |
| **Prevalence of *Plasmodium* species** | Microscopy | | | |
| | *P. falciparum* | 57.3 (59) | 42.7 (44) | 66.0 (103) |
| | *P. malariae* | 63.4 (26) | 36.6 (15) | 26.3 (41) |
| | *P.f/P.m* | 83.3 (10) | 16.7 (02) | 7.7 (12) |
| | PCR | | | |
| | *P. falciparum* | 64.3 (146) | 35.7 (81) | 81.9 (227) |
| | *P. malariae* | 100 (01) | 0.0 (0) | 0.4 (01) |
| | *P.f/P.m* | 54.2 (26) | 45.8 (22) | 17.3 (48) |
| | *P.f/P.m/P.o* | 100 (01) | 0.0 (0) | 0.4 (01) |
| **Geometric mean trophozoite (trophozoite/μL** | *P. falciparum* | 1515 | 1201.3 | 1358.2 |
| | *P. malariae* | 1020 | 1091.8 | 1055.9 |
| **Geometric mean gametocyte (gametocyte/uL)** | *P. falciparum* | 5.3 | 13.3 | 9.3 |
| | *P. malariae* | 2.7 | 0 | 1.4 |
| **Have you heard of malaria?** | Yes | 30.8 (74) | 69.2 (166) | 48.5 (240) |
| | No | 92.5 (236) | 7.5 (19) | 51.5 (255) |
| **Source of information about malaria?** | Radio/TV | 0.0 (0) | 100 (03) | 0.6 (03) |
| | School | 28.7 (56) | 71.3 (139) | 39.4 (195) |
| | CHW | 46.7 (14) | 53.3 (16) | 6.1 (30) |
| | Sch/CHW | 0.0 (0) | 100 (03) | 0.6 (03) |
| | Radio/Sch | 0.0 (0) | 100 (07) | 1.4 (07) |
| | Nil | 93.0 (240) | 7.0 (18) | 52.1 (258) |
| **Do you own an LLIN?** | Yes | 58.1 (225) | 41.9 (162) | 78.2 (387) |
| | No | 78.7 (85) | 21.3 (23) | 21.8 (108) |
| **Do you sleep under a LLIN?** | Yes | 58.1 (137) | 41.9 (99) | 47.7 (236) |
| | No | 66.8 (173) | 33.2 (86) | 52.3 (259) |
| **How frequent do you sleep under a LLIN?** | Always | 65.8 (73) | 34.2 (38) | 22.4 (111) |
| | Often | 52.2 (59) | 47.8 (54) | 22.8 (113) |
| | Rarely | 41.7 (05) | 58.3 (07) | 2.4 (12) |
| | Never | 66.8 (173) | 33.2 (86) | 52.3 (259) |
| **Anti-malaria intake last 30days** | Yes | 23.1 (06) | 76.9 (20) | 5.3 (26) |
| | No | 64.8 (304) | 35.2 (165) | 94.7 (469) |
| **What is the cause of malaria?** | Mosquito bite | 27.2 (49) | 72.8 (131) | 36.4 (180) |
| | Dirty water | 37.5 (12) | 62.5 (20) | 6.5 (32) |
| | Sour food | 50.0 (04) | 50.0 (04) | 1.6 (08) |
| | Swimming | 0.0 (00) | 100 (02) | 0.4 (02) |
| | Barefoot | 100 (01) | 0.0 (00) | 0.2 (01) |
| | Mosq/DW | 16.7 (01) | 83.3 (05) | 1.2 (06) |
| | Mosq/SF | 0.0 (00) | 100 (02) | 0.4 (02) |
| | Mosq/SF/DW | 0.0 (00) | 100 (01) | 0.2 (01) |
| | No idea | 92.4 (244) | 7.6 (20) | 53.3 (264) |

Key: Sch = School, CHW = Community Health Worker, Mosq = Mosquito bite, DW = Dirty water, SF = Sour Food; LLIN = Long-lasting Insecticide Treated Net

($\chi^2$ = 30.5, p < 0.0001) (Fig 2). PCR assay revealed that the majority of the asymptomatic malaria infections among children were due to *P. falciparum* mono-infections accounting for a prevalence of 81.9% (227); with a substantial proportion [17.3% (48)] of *P. falciparum* + *P. malariae* mixed infection. The 3–9 age structure [*Pf*: 83.9% (146); *Pf+Pm*: 14.9% (26)] and the male sex [*Pf*: 78.8% (115); *Pf+Pm*: 20.5% (30)] recorded the highest prevalence for both species (Fig 2). While a single case of triple *Pf+Pm+Po* was observed, no infection with *P. vivax* was detected (Fig 2).

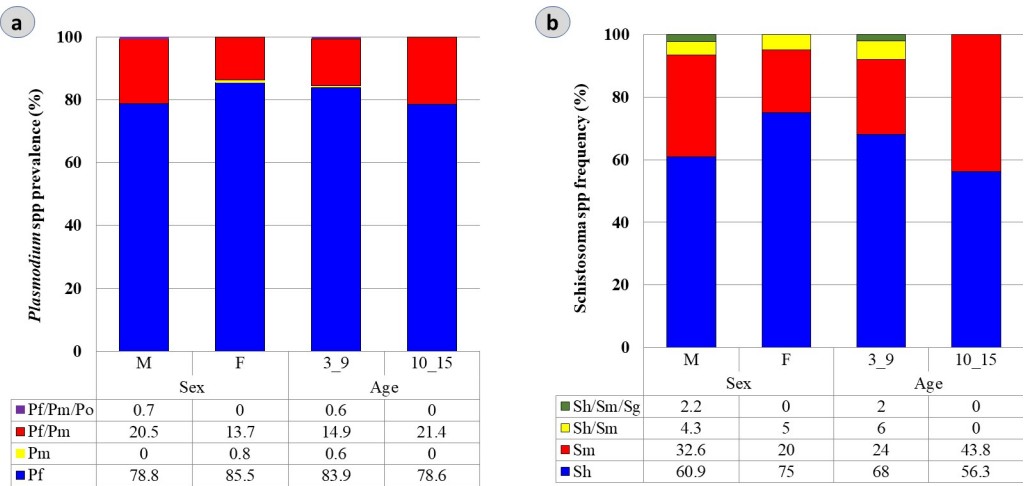

Key: Pf = *P. falciparum*; Pm = *P. malariae*; Pf/Pm = Mixed *P. falciparum* + *P. malariae* ; Pf/Pm/Po = Mixed *P. falciparum* + *P. malariae* + *P. ovale* infections
Sh = *S. hematobium*; Sm = *S. mansoni*; Sh/Sm = Mixed *S. hematobium* + *S. mansoni*; Sh/Sm/Sg = Mixed *S. hemtobium* + *S. mansoni* + *S. guineensis*

**Fig 2. Age and Sex related prevalence and species composition of (a)** *Plasmodium* **and (b) Schistosome.**

## Comparison of diagnostic accuracy between RDT and microscopy using nPCR as the reference for malaria parasite detection

The sensitivity, specificity, and positive and negative predictive values of microscopy and RDT against PCR are presented in Table 3. Using the highly sensitive PCR as the reference test, 314/ 495 participants were positive for malaria. Out of the 314 positive samples, RDT identified 62.3% (202) infections while microscopy detected 36.3% (114) true positives ($\chi^2$ = 49.3, p < 0.0001) (S1 & S2 Figs). However, RDT and microcopy each recorded 3.2% (16) and 8.3% (41) false positives respectively ($\chi^2$ = 12.1, p = 0.0005). Microscopy reported a higher false negative rate (42.2%), missing out on 209 PCR positive samples than RDT which had a false

**Table 3. Test performance of the different diagnostic assays used for evaluating malaria positivity from blood samples.**

| | | nPCR | | |
|---|---|---|---|---|
| | | Positive | Negative | Total |
| **Microscopy** | Positive | 114 | 41 | 155 |
| | Negative | 209 | 131 | 340 |
| | Total | 323 | 172 | 495 |
| | | nPCR | | |
| | | Positive | Negative | Total |
| **RDT** | Positive | 202 | 16 | 218 |
| | Negative | 122 | 155 | 277 |
| | Total | 324 | 171 | 495 |

| | nPCR | | | | | |
|---|---|---|---|---|---|---|
| **Diagnostic tool** | **Parameters (95% CI)** | | | | | |
| | Sensitivity | Specificity | PPV | NPV | Accuracy | Kappa |
| **Microscopy** | 35.3% [32.2% - 37.6%] | 76.2% [73.5% - 79.9%] | 73.5% [71% - 76.2%] | 38.5% [35.6% - 42.2%] | 49.5% [47.2% - 52%] | 0.09 |
| **RDTs** | 62.3% [59.9% - 64.3%] | 90.6% [88.1% - 93.1%] | 92.7% [89.3% - 95.9%] | 55.9% [53.2% - 58.6%] | 72.1% [69.5% - 74.5%] | 0.51 |

negative rate of 24.6% failing to detect 122 PCR positive samples ($\chi^2$ = 48.4, p < 0.0001). Generally, RDT had a 2-fold more sensitive score than microscopy (3.5% vs 1.8%) with the sensitivity of microscopy and RDT being 35.3% and 62.3% respectively. Also, RDT reported higher specificity (90.6% vs 76.2%), PPV (92.7% vs 73.5%), and NPV (55.9% vs 38.5%) than microscopy. Cohen's kappa statistic revealed moderate agreement (k = 0.5) between RDT and PCR and slight agreement (k = 0.1) between microscopy and PCR. Generally, PCR and RDTs recorded a higher sensitivity than microscopy in malaria diagnosis during the dry season. A high prevalence of 42.2% sub-microscopic infection carriage was recorded in this study (Table 3; S1 & S2 Figs).

## Association between KAP, LLINs ownership, LLINs use, and malaria prevalence

There was no significant difference in malaria prevalence between children who reported good KAP [$\chi^2$ = 0.8; p = 0.37] and those who did not. Similarly, children who owned and slept under insecticide-treated nets had a 1.1% lower *Plasmodium* infection prevalence than their counterparts who did not possess nor sleep under LLINs [$\chi^2$ = 2.6; p = 0.1] as shown in Table 4. Additionally, a 1.5% reduction in malaria prevalence was observed among children who owned and slept under a LLIN compared to children who owned a LLIN without sleeping under.

## The prevalence of schistosomiasis, geohelminthiasis, and the intensity of infection

In this study, 13.3% (66/495) of participants were positive for schistosomiasis comprising 9.5% (47/495) urogenital and 3.8% (19/495) intestinal forms. Generally, males [69.7% (46/66)] documented significantly higher infection prevalence than females [30.3% (20/66)] ($\chi^2$ = 11.5, p = 0.0007). Likewise, the 3–9 years category [75.8% (50/66)] was more remarkably infected than the 10–15 years [24.2% (16/66)] ($\chi^2$ = 16.8, p < 0.0001) (Fig 2). The 3–9 years age group recorded a high prevalence of 72.3% (34/47) *S. haematobium*-only infection compared to 19.1% (9/47) of the 10–15 group in the urogenital schistosomiasis cohort ($\chi^2$ = 29.1, p < 0.0001) as seen in Table 5. Males (59.6%, 28/47) were more infected than females (31.9%, 15/47) ($\chi^2$ = 15.1, p = 0.0003). Similarly, for intestinal schistosomiasis, *S. mansoni* egg infection rate was higher in the 3–9 years range [78.9% (15/19)] and the male gender [78.9% (15/19)].

**Table 4. KAP and LLINs predictors of malaria prevalence.**

| Predictor | Malaria outcome | Malaria prevalence (PCR); % (n) | | Statistic |
| | | Age groups (yr) | | |
| | | 3–9 | 10–15 | $\chi^2$ [p-value] |
|---|---|---|---|---|
| KAP (+) | Positive | 63.5 (33) | 70.3 (90) | 0.80 [0.37] |
| | Negative | 36.5 (19) | 29.7 (38) | |
| KAP (-) | Positive | 59.5 (141) | 61.9 (13) | 0.05 [0.83] |
| | Negative | 40.5 (96) | 38.1 (8) | |
| LLINs ownership (+) & LLINs use (+) | Positive | 25.6 (34) | 23 (23) | 17.1 [0.03*] |
| | Negative | 74.4 (99) | 77 (77) | |
| LLINs ownership (-) & LLINs use (-) | Positive | 55.3 (47) | 73.9 (17) | 2.59 [0.1] |
| | Negative | 44.7 (38) | 26.1 (6) | |
| LLINs ownership (+) & LLINs use (-) | Positive | 58.2 (53) | 69.8 (44) | 2.15 [0.14] |
| | Negative | 41.8 (38) | 30.2 (19) | |

**Table 5. Prevalence and intensity of *Schistosoma* spp and STHs infections.**

| Category | | Age (yr) | | Total |
|---|---|---|---|---|
| **Variables** | | **3–9** | **10–15** | |
| **% (n)** | | 62.6 (310) | 37.4 (185) | 100 (495) |
| **Schistosomiasis prevalence** | | 75.8 (50) | 24.2 (16) | 13.3 (66) |
| **Prevalence of *Schistosoma* species** | *S. haematobium* | 79.1 (34) | 20.9 (09) | 65.2 (43) |
| | *S. mansoni* | 63.2 (12) | 36.8 (07) | 28.8 (19) |
| | *S.h/S.m*[1] | 100 (03) | 0.0 (00) | 4.5 (03) |
| | *S.h/S.m/S.g*[1] | 100 (01) | 0.0 (00) | 1.5 (01) |
| **Prevalence of STHs** | *A. lumbricoides* | 70.0 (14) | 30.0 (06) | 30.3 (20) |
| | Hookworms | 66.7 (02) | 33.3 (01) | 4.5 (03) |
| | *T. trichuris* | 100 (01) | 00 | 1.5 (01) |
| ***Intensity of helminths eggs** | *S. haematobium* | 153.4 | 62.9 | 108.2 |
| | *S. mansoni* | 125.2 | 47.4 | 86.3 |
| | *S. guineensis* | 3.2 | 8.0 | 5.6 |
| | Ascaris | 381.5 | 180.7 | 281.1 |
| | Hookworm | 12 | 04 | 08 |
| | Trichuris | 04 | 00 | 02 |

Key: Sh = *S. haematobium*; Sm = *S. mansoni*; Sg = *S. guineensis*; [1]Presence of mixed *Schistosoma* species in urine. *The following WHO egg count classifications were used to determine infection intensity for: *A. lumbricoides* infection: light = 1–4,999 epg, moderate = 5,000–49,999 epg and heavy = ≥ 50,000 epg; Hookworm: light = 1–1,999 epg; moderate = 2,000–3,999 epg; and heavy = ≥4,000 epg while for *T. trichiura*: light = 1–999 epg, moderate = 1,000–9,999 epg and heavy = ≥ 10,000 epg.

Interestingly, mixed-species infection of *S. haematobium* + *S. mansoni* and *S. haematobium* + *S. mansoni* + *S. guineensis* were observed in urine samples of infected individuals at a frequency of 6.4% (3/47) and 2.1% (1/47) respectively ([Fig 2]). Gender disparity data showed that males [62.1% (41/66)] had a higher schistosome egg infection rate than females [37.9% (25/66)] with the former scoring a prevalence of 57.4% (27/47) *S. haematobium* mono-and 78.9% (15/19) *S. mansoni* mono-infections than the later [*S. haematobium*: 34.0% (16/47) and *S. mansoni*: 21.1% (4/19)]. The disparity in excretion of *S. haematobium* only eggs reveals that 27.9% (12/43) had a heavy infection (HI: ≥50 eggs/10ml of urine) while 72.1% (31/43) had a light infection (LI: <50 eggs/10ml of urine). The geometric mean load was 211.1 (range: 76–1079) and 11.0 (range: 2–48) eggs per 10ml of urine for heavy and light infections respectively. Generally, high mean egg loads were observed in males and the 3–9 years group. Egg shedding varied between females [HI = 22.2% (4/18); LI = 77.8% (14/18); mean egg count = 25.5 (range: 3–384)] and males [HI = 31.0% (9/29); LI = 68.9% (20/29); mean egg count = 34.4 (range: 2–1079)] and among the different age groups including the 3–9 years [HI = 32.4% (11/34); LI = 67.6% (23/34); mean egg count = 19.9 (range: 2–384)] and 10–15 category [HI = 22.2% (2/9); LI = 77.8% (7/9); mean egg count = 69.1 (range: 3–1079)]. Similarly, *S. mansoni* egg excretion was reported for 28.8% (19/66) infected children categorized as 10.5% (2/19) heavy; 42.1% (8/19) moderate, and 47.4% (9/19) light infections. The geometric mean egg counts were 605.9 (range: 510–720), 211.9 (range: 264–384), and 55.1 (range: 24–96) eggs per gram of stool for heavy (HI: ≥400), moderate (MI: 100–399) and light (LI: 1–99) infections respectively. Egg expulsion differed between males [HI = 13.3% (2/15); MI = 40% (6/15); LI = 46.7% (7/15); mean egg count = 128 (range: 24–1254)] and females [MI = 50% (2/4); LI = 50% (2/4); mean egg count = 80.7 (range: 24–192)] and among the 3–9 age group [HI = 20% (3/15); MI = 40% (6/15); LI = 40% (6/15); mean egg count = 102.4 (range: 24–384)] and 10–15 years cluster [MI = 25% (1/4); LI = 75% (3/4); mean egg count = 106.4 (range: 24–720)]. Variation in *S. haematobium* + *S. mansoni* infection prevalence between gender and age groups reveals that males [66.7% (2/3); MEC = 17.9% were more infected than females [33.3% (1/3);

**Table 6. Pairwise comparison between microhematuria and nucleopore urine filtration in urogenital schistosomiasis diagnosis.**

| Category: Age/Sex | | Microhematuria (MH) | | |
|---|---|---|---|---|
| | | **Positive** | **Negative** | **Total** |
| **Nucleopore Urine filtration (NUF)** | Positive | 38 | 9 | 47 |
| | Negative | 13 | 435 | 448 |
| | Total | 51 | 444 | 495 |

MEC = 6]. Similarly, only the 3–9 years [88.9% (8/9); MEC = 23.7%] documented infection. Triple mixed *S. haematobium* + *S. mansoni* + *S. guineensis* eggs were observed only in males [25% (1/4); MEC = 53.2] with no apparent difference between age groups.

There was a significant correlation (r = 0.8) between microhematuria (prevalence: 10.3%) and nucleopore urine filtration in the diagnosis of urogenital schistosomiasis with a marked difference ($\chi^2$ = 279.7, p < 0.0001) between infected (MH$^+$NUF$^+$ = 38) and un-infected (MH$^-$NUF$^-$ = 435) individuals as highlighted in Table 6. Microhematuria exhibited a sensitivity and specificity of 80.9% and 97.1%. Among the STHs, *A. lumbricoides* was the most common [4.0% (20/495); 3–9 years: 70% (14); male: 65% (13)]; Hookworms [0.6% (3/495)] and Trichuiris [0.2% (1/495)].

## Association between KAP, WaSH, Praziquantel (PZQ) intake and schistosomiasis outcome among the study participants

There was a significant association between schistosomiasis risk factors and disease outcome. The most important factors associated with infection in the study areas were age and gender. Variables such as poor knowledge, insufficient WaSH practices, and poor PZQ uptake during school-based treatment campaigns were higher in children between 3–9 years, correlating with high schistosomiasis prevalence. Similarly, higher schistosomiasis prevalence was recorded in males exhibiting high-risk factor scores including KAP, WaSH, and PZQ intake as shown in S1 Table.

Higher prevalence values correlated with lower class levels [unregistered to class (I-III): 75.8%]; poor knowledge on the cause of bilharzia (81.8%); bathing in streams (80.3%); use of streams as a daily water source (68.2%); absence of communal toilets (69.7%); open-air defecation and use of stream as excretion site (68.2%); lack of functional health club in school (78.8%); unawareness about periodic school deworming program (69.7%) and the tendency of missing PZQ administration in school by the teacher (81.8%) (Table 7).

## Prevalence of malaria and schistosome polyparasitism and impact on anemia level

The prevalence of malaria/schistosomiasis and malaria/STHs polyparasitism was 11.9% (59/495) and 1.2% (6/495) respectively with the predominance of *P. falciparum*/*S. haematobium* co-infection followed by *P. falciparum*/*S. mansoni*. The overall anemia prevalence was 32.9% (163/495) with a greater proportion categorized as moderate. A positive association (r = 0.9; p < 0.01) between *P. falciparum* prevalence and schistosome infection was observed. Children infected with *S. mansoni* had higher *P. falciparum* parasite density compared to *S. haematobium*/*Plasmodium spp* co-infection ($\chi^2$ = 10.31; p = 0.04) or *P. falciparum* mono-infection ($\chi^2$ = 4.77; p = 0.08). *P. falciparum*/*S. haematobium-infected* children had lower mean parasite density than their *P. falciparum*-only counterpart ($\chi^2$ = 2.09; p = 0.13) with a reduction in hemoglobin concentration. Similarly, mixed *P. falciparum* + *P. malariae* infection with *S. mansoni* was associated with a two-fold higher malaria parasite load than the *S. haematobium*

**Table 7. Schistosomiasis risk factors analysis with age and sex.**

| Category | Schistosomiasis risk factor score: % (n) | | | |
|---|---|---|---|---|
| | Age (years) | | Univariate analysis | Multivariate analysis |
| | 3–9 | 10–15 | cOR (95% CI); [p-value] | aOR (95% CI); [p-value] |
| Class level: unregistered to class (I-III) | 59.1 (39) | 16.7 (11) | 1.6 (0.46–5.63) [0.45] | / |
| Poor knowledge of schistosomiasis | 65.2 (43) | 21.2 (11) | 2.79 (0.74–10.50) [0.12] | 3.05 (0.87–11.13) [0.09] |
| Absence of communal pipe-borne water | 28.8 (19) | 4.5 (3) | 2.66 (0.67–10.55) [0.15] | 2.61 (0.63–10.59) [0.15] |
| Bathing in streams | 62.1 (41) | 18.2 (12) | 1.52 (0.39–5.81) [0.54] | / |
| Use of stream as a daily water source | 59.1 (39) | 9.1 (6) | 7.09 (2.16–23.23) [0.002*] | 8.11 (3.52–24.76) [0.002*] |
| Absence of communal toilets | 63.6 (42) | 6.1 (4) | 15.75 (4.03–61.42) [< 0.00001*] | 19.75 (7.22–63.87) [< 0.00001*] |
| Absence of household toilets | 25.8 (17) | 4.5 (3) | 2.23 (0.56–8.92) [0.25] | / |
| Open-air defecation and excrete in streams | 60.6 (40) | 7.6 (5) | 8.80 (2.49–31.15) [0.0003*] | 11.05 (3.99–29.37) [0.0003*] |
| Lack of functional health club in school | 65.2 (43) | 13.6 (9) | 4.78 (1.34–17.02) [0.01*] | 4.13 (1.01–13.69) [0.02*] |
| Unaware of periodic school deworming | 56.1 (37) | 13.6 (9) | 2.21 (0.69–7.15) [0.18] | 5.91 (1.37–9.22) [0.23] |
| Missed taking PZQ regularly | 62.1 (41) | 19.6 (13) | 1.05 (0.25–4.47) [0.35] | / |

Significant: *

equivalent ($\chi^2$ = 13.16; p = 0.02). Contrarily, *P. falciparum*, and *Ascaris* infection were negatively correlated although a higher mean trophozoite density compared to *P. falciparum* mono-infection ($\chi^2$ = 3.39; p = 0.08) was observed for the few co-infected cases as shown in Table 8.

**Table 8. Association between *Plasmodium* species and helminths worms and impact on mean parasite density and Hb level.**

| Category | Prevalence: % (n) | +Geometric mean parasite density (parasites/µL) | Anemia status: Hb level [% (n)] | | |
|---|---|---|---|---|---|
| | | | Age (years) | | Fisher exact test |
| | | | 3–9 | 10–15 | |
| §*Pf* only | 62.8 (174) | 1263.8 | 10.2 [46.6 (81)] | 10.4 [13.2 (23)] | < 0.00001* |
| §*Pf* + *Pm* only | 13.4 (37) | 806.1 | 9.6 [51.4 (19)] | 10.4 [27.0 (10)] | 0.0554 |
| *Pf/Sh* | 55.9 (33) | 1072.9 | 9.9 [63.6 (21)] | 10.1 [15.2 (5)] | 0.0025* |
| *Pf/Sm* | 18.6 (11) | 2249.9 | 10.3 [72.7 (8)] | 10.8 [9.1 (1)] | 0.0545 |
| *Pf/Sh + Sm* | 5.1 (3) | 160 | 10.5 [100 (3)] | 0 | 1 |
| *Pf* + *Pm/Sh* | 11.9 (7) | 415.7 | 9.6 [71.4 (5)] | 0 | 0.0476* |
| *Pf* + *Pm/Sm* | 6.8 (4) | 784.1 | 10.3 [75 (3)] | 0 | 1 |
| *Pf/Ascaris* | 30 (6) | 2287.5 | 10.2 [50 (3)] | 10.5 [16.7 (1)] | 1 |

§Prevalence of *Pf* and *Pm* was inferred from PCR; +Mean Parasite density was computed from microscopy/PCR true positives.

*Significant p < 0.05; Key: *Pf* = *P. falciparum*; *Pm* = *P. malariae*; *Sh* = *S. haematobium*; *Sm* = *S. mansoni*; Hb = Hemoglobin

## Discussion

Malaria and helminth infections are responsible for a significant burden of morbidity and mortality in children across many parts of the world. Dependent on *Anopheles* mosquito and snail intermediate host (for schistosomiasis) for complete life-cycle propagation, *Plasmodium* and schistosome parasites in particular encounter a remarkable challenge during the dry season in regions where the absence of rain limits the vector or intermediate host abundance and survival for several months [35, 36]. Moreover, the social, ecological, and environmental drivers facilitating the transmission of these diseases are significantly lessened during the dry season. While the majority of malaria and helminthiasis cases are predominant during the wet season, clinically silent asymptomatic *P. falciparum* and helminth infections can persist through the dry season and constitute an important reservoir for transmission during the advent of the next rainy season. This study, therefore, investigated the prevalence and factors driving sustained malaria and helminths transmission despite regular delivery of long-lasting insecticide-treated nets and systematic PZQ/ALB preventive chemotherapy campaigns by control programs. Whilst there has been a sizeable reduction in malaria burden among children < 5 years in the North Region of Cameroon from 2000 to 2020; [2] the current study reports a high *Plasmodium* prevalence in children notably between 5–10 years of age.

This epidemiological shift in malaria prevalence from younger preschool children (< 5 years) who constitute the primary at-risk group to school-age children (5–15 years) is attributed to the massive implementation of various control strategies that have largely contributed to protecting the under-five vulnerable populations [7]. Compared with younger children, SAC often experiences mild clinical disease, usually harboring infections that go untreated. Also, their reduced likelihood of utilizing malaria prevention methods such as insecticide-treated bed nets further predisposes them to infectious mosquito bites that facilitate transmission. This is contrary to the preschool where the scale-up of LLINs and ACTs has led to a decline in malaria prevalence in this age group although mean parasite density is often high owing to low-level acquired immunity.

Significant attention still surrounds the public health relevance of asymptomatic and submicroscopic infections due to their involvement in parasite transmission and the potential threat of compromising malaria control and elimination efforts. A high prevalence of *P. falciparum* asymptomatic malaria in the SAC group was observed at a prevalence of 46.9%, lower than the 91.6% recorded during the rainy season in this Region by a previous study [18]. These asymptomatic infection carriages are the major reservoir of silent circulating parasite biomass; especially in areas with seasonal transmission patterns. These asymptomatic infections may affect the performance of epidemiological diagnostic tools such that detection of these parasites becomes difficult and is missed by traditional microscopy and RDTs [31]. These results expand on growing evidence across Africa that SAC are significant contributors to the *P. falciparum* infection burden and should be a focus of future malaria control interventions such as school-based preventive treatment and expanded seasonal malaria chemoprevention (SMC) [6].

Males were found to have a higher *Plasmodium* infection prevalence and mean parasite density than their female counterparts probably because of the tendency to participate in outdoor activities for longer periods without protection and the non-adherent to sleeping under LLINs. Although a low gametocyte prevalence (0.8%) and density (29.8 gametocytes/μL) were observed in this study using microscopy, the prevalence is likely underestimated as submicroscopic infections have been shown to yield high gametocyte counts when a more sensitive and accurate technique like RT-qPCR is employed [31]. Nevertheless, the reduced gametocyte frequency may be due to parasites minimizing their investment in transmission to coincide with the apparent reduction in vector abundance particularly during the dry season [37].

The substantial prevalence of PCR-detected *P. malariae* co-existing with *P. falciparum* (17.3%) among asymptomatic children in the dry season is mainly due to its sensitivity in detecting low sub-microscopic parasitemia which often occurs in mixed infections and is frequently underestimated by microscopy. Moreover, the parasite's biological attributes such as its ability to undergo recrudescence [38] and a 72hrs lengthy asexual life cycle in RBCs than *P. falciparum* triggers an infection pattern mediated by a low number of merozoites per erythrocytic cycle making it difficult to capture by microscopy; as such constituting a reservoir for continuous transmission. Co-infection of *P. falciparum* and *P. malariae* in children was responsible for anemia (10.5%, 29/277) probably as a result of the double impact on RBC lysis, further contributing to the substantial morbidity burden.

On the other hand, the epidemiology and transmission of schistosomiasis in the North Region is governed by a nexus of human behavior, cultural factors, socio-economic status, and ecology cooperating to spur the biological interaction between the human and snail host life cycle stages of the parasite. This establishes the relevance of community monitoring and evaluation of the prevalence and intensity of infection to document the impact of treatment success and optimize control program outputs [27]. Schistosomiasis presents a public health challenge in this Region where a majority of the inhabitant population rely on water networks of the Geoges de Kola and Lagdo dam for daily activities [19]. This may also be influenced by the population influx from the Boko Haram conflict-hit zones which further imposes a huge reliance on water bodies for daily activities thereby facilitating disease transmission [22]. This study, therefore, highlights the sympatric transmission of persistent urogenital and intestinal schistosomiasis in three communities characterized by a high prevalence, risk, and severity of infection in the 3–9 years age cohort and males than their respective counterparts despite fifteen years of regular PZQ and ABZ school-based mass deworming campaigns. However, due to the constrained 2020 COVID pandemic wave, PZQ and ABZ MDA campaigns were not implemented in this Region. Age and gender were two key predictive determinants of infection. Children in the age groups 3–9 years were twice more likely to be at risk of schistosomiasis infection than their counterpart. This younger age group exhibits a high frequency of water contact for domestic purposes [36]. Also, since PZQ does not prevent reinfection of the immature worm stages, parasite transmission cannot be interrupted. However, besides PZQ administration, the contribution of age-acquired immunity to (re)infection accounts for a decrease in infection prevalence for the older 10–15 years children [36]. The intensity of both *S. haematobium* and *S. mansoni* egg excretion was significantly lower in the older aged children. T Intensity of egg excretion is largely influenced by age-mediated immunity to a reduction in worm fecundity and egg load [39]. This well-documented inverse relationship between prevalence, infection intensity, and increasing age could be explained by the direct effect of parasite metabolic regulation and host-mediated innate resistance to infection that drives a strong protective immunity associated with increased hormonal levels during puberty [36]. Additionally, the wide access and use of ACTs may have also contributed to the observed reduction in egg intensity [40].

As predicted, sex variation in prevalence and intensity of egg excretion was significantly higher in the male gender due to prolonged exposure to cercariaeted-snail-infested streams during swimming and fishing activities which are more common amongst males than females [41]. However, this differs from other studies reporting females as the predominantly infected gender probably owing to the frequent water contact tendencies involved during laundry activities. The observed intensity of egg infection in both males and females poses a high risk of genital schistosomiasis; amplified by the high frequency of unawareness (58.5% males; 64% females) about schistosomiasis risk factors [39]. This is further supported by the presence of *S. haematobium* eggs in urine alongside verbal reports of females experiencing hematuria,

dysuria and itchy urination. Additionally, a low frequency of mixed schistosome infections (*S. haematobium*, *S. mansoni*, and *S. guineensis)* were identified in children. This could be due to the focal distribution of the snail intermediate hosts and substantial human mobility between transmission sites [42]. Indeed, Leger & Webster (2016) previously revealed evidence of mating between *S. haematobium* males and *S. mansoni* female adult worms in Central Africa leading to the deposition and excretion of hybrid eggs through the urogenital tract [42]. Although genotyping was no performed, some of the observed *S. haematobium* eggs may actually be hybrids and consequently impact the nature of disease morbidity (e.g. anemia, liver pathology) and the success of MDA control programs [43]. Infection load, defined by egg count was generally low with 9.5% shedding eggs in urine characterized by a majority (72.3%) of light infection intensity status (< 50 eggs/10 ml urine). This could be attributed to shrinkage in the factors favoring schistosomiasis transmission and infection intensity including age, minimal water contact, the drying-up of temporal water bodies, marked reduction in snail population density during the dry season, and concomitant immunity as a consequence of past infections or polyparasitism [36]. In particular, the geographical distribution and abundance of the snail intermediate host (*Bulinus spp* or *Biomphalaria spp*) is a principal determinant, accounting to a large extent for variation in the seasonal transmission of the disease. Adult snails die during the dry season while the younger snail population aestivates under the soil to ensure adaptation and survival during the next rainy season [44]. This may explain the reason for the decreased prevalence compared to previous studies conducted during the rainy season in this region.

This study also assessed the association between schistosomiasis infection, school attendance, KAP, and water, sanitation, and hygiene (WaSH) practices among children. The observed schistosomiasis prevalence aligns with the high prevalence of children who are unenrolled and do not attend school regularly; further leading to poor therapeutic compliance outcomes. The generally observed low primary school attendance in the study localities hinders the non-school enrolled SAC and PSAC from treatment campaign benefits, thus, emphasizing the necessity to effectively include this neglected group. In line with the current WHO guidelines on schistosomiasis control and elimination [29], this study, therefore, underscores the opinion of a treatment scale-up from the school-centered approach to an intensified community-based preventive chemotherapy strategy inclusively targeting all at-risk groups (SAC, PSAC, occupation-related, pregnant women) to reduce disease morbidity [29]. Furthermore, the lack of comprehension about the factors associated with schistosomiasis transmission even after several years of school system-based PZQ campaigns reflects the limited public health knowledge about the disease. Indeed, many of the older children (9–15 years) usually accompany their parents for cattle rearing activities during school periods; further favoring the poor knowledge and reduced treatment uptake [36]. Moreover, the absence of functional health education clubs in primary schools in these communities is also contributing to the disease knowledge gap. Emphasis must be directed towards strengthening health education through the promotion and implementation of local language-based comic cartoons in schools of endemic areas as a complementary intervention for interrupting disease transmission. However, despite the fundamental role of health education in disease prevention, improvement of community water supply is imperative to limit frequent contact with streams [13]. The observed "non-symptomatic" urinary schistosomiasis in the older age group may be attributed to the combined effect of immune response and low egg infection intensity leading to possibly undetected microhematuria by urine strips which was otherwise identified by the more sensitive urine filtration technique. Importantly, the biologically significant minority hotspot of high egg shedding individuals in these localities poses concern in sustaining disease transmission necessitating further investigation on the genetic determinants predisposing such reservoir carriers.

Consistent with past research, STHs were not common in these localities [9]. Although replicate slides of a single individual fecal sample may be adequate at baseline in moderate to high endemic zones, the Kato-Katz technique may have a low sensitivity for *S. mansoni* and STHs diagnosis. In addition to mono-parasite infections, this study demonstrated the common occurrence of polyparasitism whereby 13.1% of children were simultaneously co-infected with mixed *Plasmodium* species and helminths worms. Bivariate analysis revealed a positive association between schistosomiasis and malaria possibly attributed to schistosome-mediated polarized Th2/Treg immunomodulation that leads to increased susceptibility to *Plasmodium* infection and may increase the risk of malaria transmission in areas of co-endemicity [45]. Notably, while *S. mansoni* infection was an important predictor of malaria parasite density, *S. haematobium* infection was a significant predictor of anemia and malaria prevalence. However, this correlation was negatively skewed for *Ascaris* infections probably due to the arid eco-epidemiological limitations of factors that favor the worm transmission. While previous studies have shown the protective effect of *Ascaris* worms on *P. falciparum* infection [46] this observation should be interpreted with caution as only six cases of *Ascaris/P. falciparum* prevalence was documented eventhough concomitant *Ascaris/P. falciparum* infection was associated with a high geometric mean asexual *P. falciparum* trophozoite carriage than *P. falciparum* mono-infection. This shows that the association between malaria and helminthic infections may depend on the host age and specific type of worm infection and therefore underscores further investigation [5, 46]. Both *Plasmodium* parasite and *Plasmodium*-helminth infection types were associated with moderate anemia as indicated by the mean hemoglobin score. This may probably be explained by the "light" majority of helminth infections observed supported by the fact that anemia caused by helminths is dependent on the intensity of infection [5]. In agreement with other studies, lower Hb levels and anemia correlated with single and multi-parasitic infections [46]. This observation reveals that synergistic interaction between multiple parasite infections increases the risk and clinical outcome of anemia particularly in children.

The cross-sectional design of this study is a limitation; implying that a longitudinal characterization involving a higher sample size of children in these localities will provide fine insights into the biology, interaction, and transmission dynamics of malaria, schistosomiasis, and STHs. Similarly, this study focused on quantifying schistosomiasis prevalence majorly in school-age children without including other high-risk groups including pregnant women, and occupation-related inhabitants. Also, schistosomiasis diagnosis was based on parasitological capture that employed only a single collection of urine and stool samples instead of the standard three consecutive samples and this may have underestimated the prevalence by missing patent infection. Molecular technique will be relevant for future studies on schistosomiasis in this region particularly for genotyping of the *S. haematobium* eggs to identify hybrid populations. Likewise, malacological and ecological surveys including snail population bionomics and cercariae infection in mollusks which constitute key targets in schistosomiasis control were not done. However, the current findings provide a situational prevalence of these diseases during the dry season.

## Conclusion

These findings demonstrate that despite regular LLIN distribution and over fifteen years of preventive chemotherapy campaigns, multiple parasite infections are frequent in children who constitute hidden reservoirs of asymptomatic malaria infections and epidemiological carriers of schistosome worms. This suggests that malaria control must prioritize targeting asymptomatic cases through sensitive diagnostic testing strategies and expanded treatment interventions. Likewise, interruption of schistosomiasis transmission in these villages requires multisectoral

health system strengthening efforts involving high-level PZQ/ALB geographic coverage and therapeutic compliance of all at-risk groups, unlimited access to WASH facilities, and community-driven health education programs structured towards improving behavioral change. This study also highlights that schistosomiasis enhances the susceptibility to *Plasmodium* infection and increases the risk of malaria and anemia in school children. Moreover, the study provides further evidence that optimizing helminth control interventions in children dually contribute to reducing malaria and anemia burden in endemic areas where both diseases geographically overlap.

## Supporting information

**S1 Fig. Numerical positivity comparison between microscopy, rapid diagnostic test, and PCR in the 3–9 years age group.**
(TIF)

**S2 Fig. Numerical positivity comparison between microscopy, rapid diagnostic test, and PCR in the 10–15 years age group.**
(TIF)

**S1 Table. Summary data on risk factors of malaria and schistosomiasis among children in Northern Cameroon.**
(DOCX)

**S1 File. Study questionnaire.**
(DOCX)

**S1 Raw images. Gel images of *Plasmodium* species PCR and microscopy image of *S. haematobium* eggs.**
(PDF)

## Acknowledgments

The authors express thankfulness to the school children who took part in this study and to their parents/guardians for consenting. Special thanks to the community leaders and district health workers of the various study localities for their assistance in data collection. The authors are also grateful to the fieldwork and laboratory staff members particularly Mr. Agbor Jean Pierre, Mr. Simon Daga, and Mr. Ahmadou Ahidjo for the specimen preparation and reading of slides. Also, appreciate Dr. Wirsy Frankline for the statistical analysis.

## Author Contributions

**Conceptualization:** Francis N. Nkemngo, Samuel Wanji, Charles S. Wondji.

**Data curation:** Francis N. Nkemngo, Lymen W. G. Raissa, Asongha M. Nkeng.

**Formal analysis:** Francis N. Nkemngo, Lymen W. G. Raissa, Derrick N. Nebangwa, Asongha M. Nkeng.

**Funding acquisition:** Francis N. Nkemngo.

**Investigation:** Francis N. Nkemngo, Lymen W. G. Raissa, Derrick N. Nebangwa, Asongha M. Nkeng, Alvine Kengne, Leon M. J. Mugenzi, Murielle J. Wondji, Robert A. Shey.

**Methodology:** Francis N. Nkemngo, Derrick N. Nebangwa, Asongha M. Nkeng, Alvine Kengne.

**Project administration:** Francis N. Nkemngo, Cyrille Ndo, Flobert Njiokou, Samuel Wanji, Charles S. Wondji.

**Resources:** Francis N. Nkemngo, Alvine Kengne, Yvan G. Fotso-Toguem, Robert A. Shey, Flobert Njiokou, Joanne P. Webster, Charles S. Wondji.

**Supervision:** Cyrille Ndo, Flobert Njiokou, Joanne P. Webster, Samuel Wanji, Charles S. Wondji.

**Validation:** Samuel Wanji, Charles S. Wondji.

**Writing – original draft:** Francis N. Nkemngo.

**Writing – review & editing:** Francis N. Nkemngo, Lymen W. G. Raissa, Derrick N. Nebangwa, Asongha M. Nkeng, Alvine Kengne, Leon M. J. Mugenzi, Yvan G. Fotso-Toguem, Murielle J. Wondji, Robert A. Shey, Daniel Nguiffo-Nguete, Jerome Fru-Cho, Cyrille Ndo, Flobert Njiokou, Joanne P. Webster, Samuel Wanji, Charles S. Wondji.

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
