## [Decision Letter · Decision Letter 0]

31 Jan 2023

PONE-D-22-34570Persistent co-transmission of malaria, schistosomiasis, and geohelminthiasis among 3-15 years old children during the dry season in Northern CameroonPLOS ONE

Dear Nkemngo,

Thank you for submitting your manuscript to PLOS ONE. After careful consideration, we feel that it has merit but does not fully meet PLOS ONE’s publication criteria as it currently stands. Therefore, we invite you to submit a revised version of the manuscript that addresses the points raised during the review process.

 Thank you for submitting your manuscript to PLoS ONE. After careful consideration, we felt that your manuscript requires revision, following which it can possibly be reconsidered. All the four reviewers expressed interest on your manuscript, however two have expressed major concerns and another two have raised minor issues. According to the reviewers, major concerns were related to the study lacking focus by having a very broad background, study design, data analysis, results and discussion.  According to the reviewers, the results are not very well presented for readers to be able to comprehend well the presented information. In addition, a number of other issues should be clarified and/or changed/adjusted, particularly, in the results and discussion sections. Lastly the manuscript should be subjected to a professional proofreading process.

We look forward to receiving your revised manuscript.

Kind regards,

David Zadock Munisi, Ph.D

Academic Editor

PLOS ONE

Journal Requirements:

Additional Editor Comments:

In addition to the reviewer's comments, I have a few more comments in the following sections of the manuscript.

**Methodology**

**
*Study design and sample size determination*
**

The calculated sample size was 370 participants, but you included 555 participants, why did you decide to include 555 participants and what was the basis for your choice of that number of participants?.The authors states that “As inclusion criteria, only children who had lived for at least a year in the selected villages and whose parents/guardians consented to participate were enrolled in the study. Those who declined to participate in the study or failed to provide feces or urine samples after the interview were excluded”. However, the authors should note that, an exclusion criterion is not the opposite of the inclusion criteria, it should be a criterion that you will use to exclude from participation individuals who have met all the inclusion criteria. In this case, those who declined participation were not qualified for inclusion in the first place. So, the exclusion criteria need to be rephrased.

**Administration of questionnaire and clinical assessment: **A simple structured questionnaire was used to collect information about each individual based on the following indicators: Instead of indicators, you may use the word “Variables”.

You have a questionnaire for information on malaria and schistosomiasis, no questionnaire for Soil-transmitted helminthiasis.

Devices and equipment used for taking measurement should be properly described (Brand, manufacturer’s etc). Equipment/devices such as Haemoglobinometer, Stadiometer, weighing balance etc.

**Data analysis**

Continuous variables were summarized into means and standard deviations (SD), and categorical variables reported as frequencies while percentages were used to document descriptive statistics. Are mean and SD, and frequencies not descriptive statistics? Kindly rephrase this.

Fisher exact test was employed for small size variable calculations - What does this mean, kindly rephrase to make it clear.Independent variables were subjected to univariate analysis to obtain odds ratios (cOR). Univariate is a broad term, which Univariate analysis did you employ, kindly make this clear.Furthermore, variables with a p-value < 0.25 underwent multivariate analysis for adjusted odds ratios (aOR) – Again, multivariate is a broad term, which multivariate analysis did you employ, kindly make this clear.

**Results**

**
*Baseline characteristics of the study participants*
**

“At the time of the survey, 26.2 % of females (59/225) and 25.6 % of males (69/270) were currently not enrolled in the local basic educational system, accounting for the literacy gap observed in this Region”…..But you also recruited prechoolers who were surely not registered in school, don’t you think those proportions also included the pre-schoolers? Is it correct to regard the proportion of unregistered preschoolers as accounting to the literacy gap in the region?

**General comment**

- All the tables need extensive revision, I suggest that the sex categorization should not appear column wise as it is now, it should rather appear once, row wise. This should apply for regions as well.

- For helminths infections, for intensity of infections to be meaningful, they have to be categorized as per WHO categorization. Just putting intensity of infection may be difficult for readers to comprehend.

Reviewers' comments:

Reviewer's Responses to Questions

**Comments to the Author**

1. Is the manuscript technically sound, and do the data support the conclusions?

Reviewer #1: Partly

Reviewer #2: No

Reviewer #3: Yes

Reviewer #4: Yes

2. Has the statistical analysis been performed appropriately and rigorously? 

Reviewer #1: I Don't Know

Reviewer #2: Yes

Reviewer #3: Yes

Reviewer #4: No

3. Have the authors made all data underlying the findings in their manuscript fully available?

Reviewer #1: Yes

Reviewer #2: No

Reviewer #3: Yes

Reviewer #4: Yes

4. Is the manuscript presented in an intelligible fashion and written in standard English?

Reviewer #1: No

Reviewer #2: Yes

Reviewer #3: Yes

Reviewer #4: Yes

5. Review Comments to the Author

Reviewer #1: Appropriate page numbers, as well as line number per page would have been very helpful in the reviewing process.

In this manuscript, the authors describe a community-based survey amongst 495 children of 3-15 years old, living in 3 communities in the North Region of Cameroon. Their survey has been performed in the dry season and the children have been examined for malaria, schistosomiasis and soil-transmitted helminths. Several epidemiological determinants of infection have been examined.

I have the impression that this is a copy of a student’s thesis. I find the introduction and the discussion too broad for a scientific publication. With more than 80 references there is no clear research focus. The authors seem to be more familiar with malaria than with helminth infections. For example: for schistosomiasis and soil-transmitted helminths (STH) most studies do focus on school aged and pre-school aged children, so I do not understand the statement on page 13 (pdf, page 8(?)) of the manuscript.

I agree with the authors that co-infection of different parasitic diseases needs more attention. The final conclusion that polyparasitism with malaria and helminth infections are still common, despite intervention via the distribution of bed nets and regular MDA treatment rounds with anti-helminthic drugs, is a relevant finding. But I have strong doubts about the scientific contribution of a study of this design, this size and this diagnostic quality. Follow-up studies of a larger cohort, preferably with different intervention arms, would have been far more informative.

The sample size calculation is based on a Schistosoma prevalence of 38.5% (page 15 of the pdf, page 8??), but it does not take into account the enormous amount of sub-analyses performed. The actual prevalence of Schistosoma infections was 13.3%, but this is based on microscopy on a single stool sample and a single urine samples. The authors do acknowledge the limitations of their diagnostics procedures for the helminths in the discussion, even repeatedly, but no explanation is given for why they did not use repeated microscopy or otherwise PCR. The use of schistosome circulating antigens is not mentioned. The lack of diagnostic sensitivity for the helminths is not in balance with the malaria.

For malaria, besides a rapid diagnostic test, also a nested PCR has been used. These procedures, as expected, detected a substantial number of additional cases. However, concerning the PCR, important details of the used laboratory procedures are lacking, only references are given. How do we know the specificity has been well controlled? Because the performed nested PCR is not a closed real-time system, this is highly relevant. Do the authors have any quality control measures in place within their laboratory? Has any internal control been used to control for potential inhibition factors.

I find the results concerning the epidemiological risk factors far too detailed and with a lot of repetition in the text of what is already depicted in the tables.

Most references are depicted with the first author’s name only. I find that not very helpful. Several key references on schistosomiasis and soil-transmitted helminths are lacking. On the other hand, I find 83 references far too much for an original research paper.

Finally: I have strong doubts whether all co-authors have seen and approved this manuscript. The name of Professor Joanne Webster has been misspelled, both in the list of authors and page 46 of the pdf (page 6???).

So, in conclusion, if this manuscript could be rewritten as a short report with much more focus on the diagnostic findings, leaving out all the epidemiological analysis and without over-interpretation of the data, I would reconsider publication.

Reviewer #2: The manuscript is well-organized and easy to follow; however, some concerns need to be addressed.

-The last sentence in the Abstract section has to be corrected both for grammar and clarity. Also, correct the grammatical errors in the “competing interests” AND “ethics statement”.

-In “study design and sample size determination”: “Parents and their respective children were invited by local community leaders to assemble at a major focal point in each of the selected communities for sample collection”. Were all children in the region recruited to participate in the study OR there was a specific sampling method? How were the communities and children selected? What happened to parents who came with more than 1 child? This has not been clearly shown in the Methods’ section.

-In “study design and sample size determination”: you included 555 participants to anticipate factors like effect size, voluntary withdrawal and for greater precision. How did you reach to this conclusion as the calculated sample size was 370? Your sampling method used, was to determine the kind of adjustment to counter the factors you mentioned. Please describe the Recruitment strategy that was employed. And counter check if the formula used belongs to Lorenz.

-In “study design and sample size determination”: “The sample size of the study population was calculated based on a previous schistosomiasis prevalence of 38.5% during the rainy season considering that malaria is endemic in this region” the statement needs citation(s).

-One of the socio-demographic data collected was a NAME. What about anonymity? Or what was the relevance of taking participants’ names instead of using unique ID numbers.

-Please indicate clearly about who answered the questionnaire. You had a group of children ranging from 3-15 years old and their guardians/parents. “By signing the form, the participants agreed to answer a questionnaire and to provide finger-prick blood, urine and stool samples for parasitological analysis”: who are the participants you are talking about here? Please clarify this.

-In sample collection and parasitological examination : “while the nested primers are specific to the 04 human Plasmodium species..” put the 04 in wordings. Please write more description on nested PCR as how you did with the primary PCR.

-For better flow in Materials and Methods,

Data collection

Questionaire

-

Clinical assessment

-

Parasitological examination of blood samples

-

Parasitological examination of urine samples

-

Parasitological examination of stool samples

-

Then provide the detailed procedures in each subsection

-In “Definitions of endpoints” : Please clarify where to expect S. mansoni and S. haematobium eggs. Is it in urine and/or stool? Because the last statement in the first paragraph is not clear. Define symptomatic urinary and intestinal schistosomiasis, separately.

-In “Results” section: You report that there was no significant difference in malaria prevalence between children who reported good KAP and those who did not. Population of your interest was children aged between 3-15, how were you able to assess KAP among these children, especially the younger ones of 3 years? This question is highly linked to a previous question as to “who answered your questionnaires”. Also, describe how you scored the children with good KAP and bad KAP under the Methods section. How many items were tested and what was the decisive score.

-“There was a significant match between the determinants of risk factors of schistosomiasis and disease outcome”. Please clarify this sentence. Do the same for the statement “…and frequent absence of consuming PZQ during school-based treatment campaigns.”

-Check numbering of the pages. There is repetition of page numbers.

-There are some abbreviations that are not defined in the document. I believe it is appropriate that the first time you wish to use an abbreviation; it has to consist a long form. Then the subsequent time, you can use it.

Reviewer #3: The research is original work of the authors and reflects an epidemiological problem of multiple infections especially in the tropics where many parasites are prevalent. The social economical information from this country could also mirror other African countries that have the same problem thus, publication of these results will benefit other control programmes. Attached please find some comments to enrich the manuscript.

After addressing the comments, the manuscript can proceed for publication.

Reviewer #4: Nkemngo et al. report in their publication the results of a cross-sectional investigation of the co-infection of malaria and Schistosoma parasites, as well as geohelminths in children.

The subject is well introduced and the methodology well described. The authors report very high levels of parasitized children during the dry season in the specific rural area, which is interesting and should communicated. It would be highly interesting to compare the results between the dry season and the rainy season in that area.

The publication uses a broad methodology to investigate the prevalence of the disease. Agreement between the different diagnostic methods is relatively low, which is rather surprising especially for the slide-positive results and PCR-negative results. Representation of the data and the statistics would be easier by using a 2x2 table.

There is something wring in the statistical analysis in the section “Association between KAP, LLINs ownership, LLINs use and malaria Prevalence” – this chapter has to be carefully checked again. In the text, the authors speak about a reduction in “%”, whereas the talk about OR in the respective table. This needs careful revision by a statistician.

Table 6 is not clear what it is supposed to show.

Minor comments:

Methods:

In the study site description, the authors should add some information on the average rainfall during the season of the collection of samples in mean and specifically for the year of collection.

Using the described malaria slide reading method – what is the obtained limit of detection for the read?

Results:

Scholl enrolment is low in the young group – but at what age school is compulsory? Shouldn’t it start at 6 years of age rather than at three years of age?

6. PLOS authors have the option to publish the peer review history of their article (what does this mean?). If published, this will include your full peer review and any attached files.

Reviewer #1: No

Reviewer #2: No

Reviewer #3: **Yes: **Jimmy Hussein Kihara

Reviewer #4: No

---

## [Author Response · Author response to Decision Letter 0]

26 Apr 2023

Responses to Editor and Reviewers’ Comments and Concerns

Title: Persistent co-transmission of malaria, schistosomiasis, and geohelminthiasis among 3-15 years old children during the dry season in Northern Cameroon [PONE-D-22-34570]

Change of title: The title of the MS has been slightly modified to reflect the data therein and now reads “Epidemiology of malaria, schistosomiasis, and geohelminthiasis amongst children 3-15 years of age during the dry season in Northern Cameroon”. 

Response to Editors’ comments

Question 1: Methodology

Study design and sample size determination

• The calculated sample size was 370 participants, but you included 555 participants, why did you decide to include 555 participants and what was the basis for your choice of that number of participants?.

• The authors states that “As inclusion criteria, only children who had lived for at least a year in the selected villages and whose parents/guardians consented to participate were enrolled in the study. Those who declined to participate in the study or failed to provide feces or urine samples after the interview were excluded”. However, the authors should note that, an exclusion criterion is not the opposite of the inclusion criteria; it should be a criterion that you will use to exclude from participation individuals who have met all the inclusion criteria. In this case, those who declined participation were not qualified for inclusion in the first place. So, the exclusion criteria need to be rephrased.

Response 1

• This research was conducted during the period of the COVID-19 pandemic with anticipation of low population turn-out due to community hesitancy linked to COVID-related myths and misinformation which we predicted could impact the overall participation. The calculated 370 was the minimum sample size. However, with the anticipated rate of compliance and study enrollment in different localities, a 50% (n = 185) sample was added, which then translated to an upper sample size of 555. This was performed to accommodate for potential attrition/loss of sample due to the inability of all the children to provide capillary blood or urine/stool samples at the time of specimen collection. Fortunately, there was a massive population turn-out in the field and we could only consider 495 participants who readily provided all three samples for analysis.

• Thank you for this intriguing remark. This has been corrected in the manuscript text. Kindly see line 214.

Question 2

• Administration of questionnaire and clinical assessment: A simple structured questionnaire was used to collect information about each individual based on the following indicators: Instead of indicators, you may use the word “Variables”.

• You have a questionnaire for information on malaria and schistosomiasis, no questionnaire for Soil-transmitted helminthiasis.

• Devices and equipment used for taking measurement should be properly described (Brand, manufacturer’s etc). Equipment/devices such as Haemoglobinometer, Stadiometer, weighing balance etc.

Response 2

• Agreed and amended

• The questionnaire also profiled information related to soil-transmitted helminthiasis, and this has now been clarified within our revised text (please see questionnaire in the supplementary file (S2File)).

• Thank you. This has been corrected in the text.

Question 3: Data analysis

• Continuous variables were summarized into means and standard deviations (SD), and categorical variables reported as frequencies while percentages were used to document descriptive statistics. Are mean and SD, and frequencies not descriptive statistics? Kindly rephrase this.

• Fisher exact test was employed for small size variable calculations - What does this mean, kindly rephrase to make it clear.

• Independent variables were subjected to univariate analysis to obtain odds ratios (cOR). Univariate is a broad term, which Univariate analysis did you employ, kindly make this clear.

• Furthermore, variables with a p-value < 0.25 underwent multivariate analysis for adjusted odds ratios (aOR) – Again, multivariate is a broad term, which multivariate analysis did you employ, kindly make this clear.

Response 3

• Thank you for this astute remark on descriptive statistics – we fully agree and the revised text has been amended accordingly (lines 306-308).

• The comment on Fisher exact test has been corrected in the text (line 311).

• Thank you for clarifying the univariate analysis term. This has been properly adjusted (line 313).

• All the possible variables selected with P < 0.25 underwent stepwise logistic regression analysis to neutralize confounding factors , and this has again been clarified within our revised text (line 314)

Question 4: Results

Baseline characteristics of the study participants

“At the time of the survey, 26.2 % of females (59/225) and 25.6 % of males (69/270) were currently not enrolled in the local basic educational system, accounting for the literacy gap observed in this Region”…..But you also recruited preschoolers who were surely not registered in school, don’t you think those proportions also included the pre-schoolers?

Is it correct to regard the proportion of unregistered preschoolers as accounting to the literacy gap in the region?

Response 4

• This is an interesting point – as technically one could say true, but equally it may be inappropriate to categorize children too young to be in school with those eligible to be in school but not attending/registered. However, for simplicity, we have updated our table and text accordingly to combine the two (with caveats acknowledged) (line 358 and Table 1)

General comment

- All the tables need extensive revision, I suggest that the sex categorization should not appear column wise as it is now, it should rather appear once, row wise. This should apply for regions as well.

- For helminths infections, for intensity of infections to be meaningful, they have to be categorized as per WHO categorization. Just putting intensity of infection may be difficult for readers to comprehend.

Response: Thank you for this comment. The tables have been extensively revised and the WHO classification score for intensity of infection has been added (lines 500-502). However, given current renewed interest by WHO and beyond in actual infection intensities, rather than the more restrictive categories, we have included both values here.

Response to Reviewers’ comments

Reviewer #1: Appropriate page numbers, as well as line number per page would have been very helpful in the reviewing process.

Agreed and amended (we apologize for not incorporating these within our original submission).

Comment 1:

In this manuscript, the authors describe a community-based survey amongst 495 children of 3-15 years old, living in 3 communities in the North Region of Cameroon. Their survey has been performed in the dry season and the children have been examined for malaria, schistosomiasis and soil-transmitted helminths. Several epidemiological determinants of infection have been examined.

I have the impression that this is a copy of a student’s thesis. I find the introduction and the discussion too broad for a scientific publication. With more than 80 references there is no clear research focus. The authors seem to be more familiar with malaria than with helminth infections. For example: for schistosomiasis and soil-transmitted helminths

(STH) most studies do focus on school aged and pre-school aged children, so I do not understand the statement on page 13 (pdf, page 8(?)) of the manuscript.

Response 1: We thank the reviewer for the manuscript synopsis and the relevant comments to improve the content of the paper. This is not a student thesis, but instead an original manuscript. The introduction and discussion were broad in an effort to provide a foundational basis for the current study. However, we have now constructively streamlined our revised manuscript for clarity and focus, and likewise reduced the number of references cited. The authors have complementary expertise on malaria and helminths with publication track record evidence in these research areas. However, the diagnostic limitation on the helminths aspect of this study is mainly due to the available resources. The statement has been corrected (line 193) and thank you for the keen observation. 

Comment 2: I agree with the authors that co-infection of different parasitic diseases needs more attention. The final conclusion that polyparasitism with malaria and helminth infections are still common, despite intervention via the distribution of bed nets and regular MDA treatment rounds with anti-helminthic drugs, is a relevant finding. But I have strong doubts about the scientific contribution of a study of this design, this size and this diagnostic quality. Follow-up studies of a larger cohort, preferably with different intervention arms, would have been far more informative.

Response 2: Whilst larger follow-up studies ideally with specific and contrasting intervention arms are always necessary and welcome. However, we respectfully disagree with the reviewer on the doubts about the scientific contribution of this study. As stated in our original manuscript, the study employed a cross-sectional design with an appropriate sample size (495) to provide preliminary/baseline data on the situation of malaria and helminths (schistosomiasis and STH) prevalence after systematic implementation of long-lasting nets, antimalarial drugs and Praziquantel/albendazole-based programmatic annual mass drug administration. Data generated from this study is vital to reinforce policy decisions. Indeed, as you correctly stated, we have in mind to conduct longitudinal follow-up studies with additional sample size to monitor the seasonal dynamics and finely characterize the drivers of persistent transmission.

Comment 3: The sample size calculation is based on a Schistosoma prevalence of 38.5% (page 15 of the pdf, page 8??), but it does not take into account the enormous amount of subanalyses performed. The actual prevalence of Schistosoma infections was 13.3%, but this is based on microscopy on a single stool sample and a single urine samples. The authors do acknowledge the limitations of their diagnostics procedures for the helminths in the discussion, even repeatedly, but no explanation is given for why they did not use repeated microscopy or otherwise PCR. The use of schistosome circulating antigens is not mentioned. The lack of diagnostic sensitivity for the helminths is not in balance with the malaria.

Response 3: The initial sample size calculation was inferred from a schistosomiasis prevalence data obtained from the National Schistosomiasis & STH Control Program. The actual prevalence (13.3%) is just a single time point and more so during the dry season (where factors governing transmission are significantly lessened due to intense drought e.g drying-up of snail-infested streams). This study was funded by a small grant with constraints in resources. However, further studies are under consideration to employ sensitive tools like PCR for actual prevalence (which may be even higher). Nonetheless, the morphological characteristic of the schistosome eggs is also a well-established diagnostic criterion used in many epidemiological studies. The CCA/CAA test kits were not available for subsidized purchase from the National Schistosomiasis Program at the time of this study because these kits are expensive and limited to small number of research laboratories (particularly the CAA kit). We agree on the point raised about the minimal (not lack) diagnostic test systems for helminths in this study. That is what was present at the time of the study and certainly, we will employ circulating antigen kits and PCR in future studies. In addition, while we agree on your point, we feel that repeated microscopy would have been much more informative for routine monitoring and evaluation of the efficacy of MDA which was not the main objective of the study (rather, the study was focused on providing baseline prevalence in the dry season). We discuss further the inherent logistical limitations of the current study within our revised text.

Comment 4: For malaria, besides a rapid diagnostic test, also a nested PCR has been used. These procedures, as expected, detected a substantial number of additional cases. However, concerning the PCR, important details of the used laboratory procedures are lacking, only references are given. How do we know the specificity has been well controlled? Because the performed nested PCR is not a closed real-time system, this is highly relevant. Do the authors have any quality control measures in place within their laboratory? Has any internal control been used to control for potential inhibition factors.

Response 4: Nested PCR is a common and frequently used method to identify Plasmodium parasites. In order to avoid redundancy, we only indicated the reference. We have now added the details of the Plasmodium speciation nested PCR method (lines 238 – 250). We used the extracted DNA of the PF3D7 strain as the laboratory positive control to avoid any discrepancy in the PCR results. Please see previous data on this from our lab (PMIDs: 36698132 & 35183684). 

Comment 5: I find the results concerning the epidemiological risk factors far too detailed and with a lot of repetition in the text of what is already depicted in the tables.

Response 5: Thank you for this remark. The data in the table has been simplified and significantly adjusted to avoid repetition

Comment 6: Most references are depicted with the first author’s name only. I find that not very helpful. Several key references on schistosomiasis and soil-transmitted helminths are lacking.

On the other hand, I find 83 references far too much for an original research paper.

Response 6: The referencing system has been corrected following the PLoS guidelines. We have streamlined and balanced the number and range of references cited within our revise text. replaced.

Comment 7: Finally: I have strong doubts whether all co-authors have seen and approved this manuscript. The name of Professor Joanne Webster has been misspelled, both in the list of authors and page 46 of the pdf (page 6???)

Response 7: Thank you for keenly picking up the spelling mistake and this has been corrected throughout the text. In the contrary, all co-authors particularly the experienced senior authors (FN, JPW, SW, CSW) read the manuscript in detail, provided critical comments and approved it for submission.

Comment 8: So, in conclusion, if this manuscript could be rewritten as a short report with much more focus on the diagnostic findings, leaving out all the epidemiological analysis and without over-interpretation of the data, I would reconsider publication.

Response 8: We sincerely thank the reviewer for the time taking to critically assess and provide detail comments and corrections on the manuscript. The manuscript has now been extensively revised to avoid redundancy and over-interpretation of the data. We hope you will find it significantly improved and re-consider it for publication.

Reviewer #2: The manuscript is well-organized and easy to follow; however, some concerns need to be addressed.

Response: Thank you for the positive remark and the comments to improve the quality of the manuscript

Comment 1:

The last sentence in the Abstract section has to be corrected both for grammar and clarity. Also, correct the grammatical errors in the “competing interests” AND “ethics statement”.

In “study design and sample size determination”: “Parents and their respective children were invited by local community leaders to assemble at a major focal point in each of the selected communities for sample collection”. Were all children in the region recruited to participate in the study OR there was a specific sampling method? How were the

communities and children selected? What happened to parents who came with more than 1 child? This has not been clearly shown in the Methods’ section.

Response 1: Thank you - last sentence in the Abstract section has been corrected accordingly (line 59). We appreciate the reviewer for this question. Recruitment followed a random sampling where children in the communities were invited in the study following the inclusion criteria. Communities were selected based on previous studies (PMID: 31139663 &14641846) and updated prevalence records from both the National Malaria Control and Schistosomiasis Control Programs. Children were selected based on age ((3 -15 years), duration of ≥ 1year in the communities, absence of severe health condition and guardian/parent consent of participation. Parents who brought more than one child were registered under the same household (or different household if the child was not theirs) once they fulfill the inclusion criteria. This has been further clarified in our revised methods section (line 212-213).

Comment 2: In “study design and sample size determination”: you included 555 participants to anticipate factors like effect size, voluntary withdrawal and for greater precision. How did you reach to this conclusion as the calculated sample size was 370? Your sampling method used, was to determine the kind of adjustment to counter the factors you mentioned. Please describe the Recruitment strategy that was employed. And counter check if the formula used belongs to Lorenz.

Response 2: We thank the reviewer for this pertinent comment, also highlighted by the editor. This study was conducted during the period of the COVID-pandemic and so we anticipated a low compliance rate and study enrollment in the different localities. In order to obtain an optimal sample size, a 50% (n = 185) sample was added to the actual calculated sample size (370) to give 555 participants. Moreover, this additive sample size was calculated to accommodate for loss of sample due to the inability of all the children to provide capillary blood or urine/stool samples at the time of specimen collection and voluntary withdrawal of participants; and that is why a random sampling strategy was employed (see Response 1) to overcome these anticipated limitations. The sentence mentioning the Lorenz formula has been modified (line 197) and the formula properly referenced. Thanks for this detail comment.

Comment 3: In “study design and sample size determination”: “The sample size of the study population was calculated based on a previous schistosomiasis prevalence of 38.5% during the rainy season considering that malaria is endemic in this region” the statement needs citation(s)

Response 3: We thank the reviewer for highlighting this citation omission. This has been effected (line 196).

Comment 4: One of the socio-demographic data collected was a NAME. What about anonymity? Or what was the relevance of taking participants’ names instead of using unique ID numbers

Response 4: Each participant’s information was anonymized using a unique code. However, the names were relevant for allocation of results and treatment administration.

Comment 5: Please indicate clearly about who answered the questionnaire. You had a group of children ranging from 3-15 years old and their guardians/parents. “By signing the form, the participants agreed to answer a questionnaire and to provide finger-prick blood, urine and stool samples for parasitological analysis”: who are the participants you are talking about here? Please clarify this.

Response 5: Since the questionnaire was orally translated in their local language (Fulfude) by a community health worker, the guardians/parents assisted the ≤ 5 years old children (n = 86) in responding to the questionnaire while > 5 years of age responded to the questionnaire themselves. This has been clarified in the text (line 151).

Comment 6: In sample collection and parasitological examination : “while the nested primers are specific to the 04 human Plasmodium species..” put the 04 in wordings. Please write more description on nested PCR as how you did with the primary PCR.

Response 6: The corrections have been made (lines 256 – 268)

Comment 7:

For better flow in Materials and Methods,

Data collection

Questionaire

Clinical assessment

Parasitological examination of blood samples

Parasitological examination of urine samples

Parasitological examination of stool samples

Then provide the detailed procedures in each subsection

Response 7: This has been modified for clarity as suggested by the reviewer. Please see the marked-up version.

Comment 8: In “Definitions of endpoints” : Please clarify where to expect S. mansoni and S. haematobium eggs. Is it in urine and/or stool? Because the last statement in the first paragraph is not clear. Define symptomatic urinary and intestinal schistosomiasis, separately.

Response 8: Thank you for the comment. This has been corrected (lines 326 to 331).

Comment 9: In “Results” section: You report that there was no significant difference in malaria prevalence between children who reported good KAP and those who did not. Population of your interest was children aged between 3-15, how were you able to assess KAP among these children, especially the younger ones of 3 years? This question is highly linked to a previous question as to “who answered your questionnaires”. Also, describe how you scored the children with good KAP and bad KAP under the Methods section. How many items were tested and what was the decisive score

Response 9: Thanks for this remark. As previously stated, children in the 3 years category were aided to respond to the question by their respective guardian/parents. The 3year old children (n = 31) exhibited poor KAP knowledge and a total of thirty-seven variables were assessed with scores between Yes or No and 0 to 1 (please S2 file). These scores were computed to give the total counts of the response to each variable. The KAP scoring system has been included in the text (lines 223 – 226).

Comment 10: “There was a significant match between the determinants of risk factors of schistosomiasis and disease outcome”. Please clarify this sentence. Do the same for the statement

“…and frequent absence of consuming PZQ during school-based treatment campaigns.”

Response 10: Thanks for this remark. This has been corrected (lines 519 to 521).

Comment 11: Check numbering of the pages. There is repetition of page numbers.

Response 11: We apologize for the inconsistent page numbering. This has now been corrected.

Comment 12: There are some abbreviations that are not defined in the document. I believe it is appropriate that the first time you wish to use an abbreviation; it has to consist a long form.

Then the subsequent time, you can use it.

Response 12: We have now defined first use of each abbreviation all throughout the text for clarity. We thank you for detail reading through the MS and enumerating the vital comments to improve on the scientific quality.

Reviewer #3: The research is scientifically sound and addresses some of the challenges that programmes face during control of parasitic infections in the community. How to integrate control of malaria and helminthes will be interesting. At policy level it may be feasible. Reading through some points can be addressed;

Response: We thank the reviewer for the positive commendation on the manuscript and the relevant comments to enrich the quality of the work.

Comment 1: The introduction can be reduced to address only what is relevant to the study as it is.

Response 1: The introduction has been reduced as suggested.

Comment 2: Some of the methods should be clearly documented with slightly more details, ie, the urine filtration, Kato katz and the delivery of the questionnaire. All information is available, just package well.

Response 2: Thank you – agreed and amended.

Comment 3: Where a test did not give different results, or conflicting information such as the immunological then the authors can ignore (such results may be good for a thesis)

Response 3: Thank you – agreed and amended.

Comment 4: In the results, the tables are many, highlight the most important glaring difference and avoid many analytical information. The narrative should be brief and up to the point of difference

Response 4: The tables have now been reorganized to make the narrative coherent and succinct.

Comment 5: The discussion can be reduced to be more specific to the research project and avoid repititition of what is obvious.

Response 5: The discussion has been reduced significantly with minimal repetition.

Comment 6: The references are many and probably, pick on the more recent.

Response 6: The references have been reduced to 46 with majority being the most recent studies.

Conclusion: The manuscript can be polished and proceed with the publication.

Response: Thank you for the comments. The MS has been significantly polished and we hope it can be considered for publication in the revised form.

Reviewer #4:

Nkemngo et al. report in their publication the results of a cross-sectional investigation of the co-infection of malaria and Schistosoma parasites, as well as geohelminths in children. The subject is well introduced and the methodology well described. The authors report very high levels of parasitized children during the dry season in the specific rural area, which is interesting and should communicated. It would be highly interesting to compare the results between the dry season and the rainy season in that area. The publication uses a broad methodology to investigate the prevalence of the disease. Agreement between the different diagnostic methods is relatively low, which is rather surprising especially for the slide-positive results and PCR-negative results. Representation of the data and the statistics would be easier by using a 2x2 table. There is something wrong in the statistical analysis in the section “Association between KAP, LLINs ownership, LLINs use and malaria Prevalence” – this chapter has to be carefully checked again. In the text, the authors speak about a reduction in “%”, whereas the talk about OR in the respective table. This needs careful revision by a statistician. Table 6 is not clear what it is supposed to show.

Response: 

• We thank the reviewer for the insightful comments. We agree that it would have been interesting to compare the data between the dry and rainy seasons and to this effect longitudinal sampling is in plan to study the dynamics of co-infection between malaria and helminthes among established cohorts of children. However, the current study focusing on the dry season was important to provide baseline data on both diseases within the scope of the available funding. Microscopic parasite density is often low in asymptomatic infection particularly during the dry season where sequestration is an advantage. With microscopy being a subjective diagnostic algorithm, artifacts may have been reported as parasites. In addition, cattle rearing are practiced in this Region and therefore Babesia (a blood-borne parasite transmitted by ticks) could have been microscopically misinterpreted for malaria parasite. That is why PCR was used for further confirmation.

• Here, we were more concerned at simultaneously and succinctly comparing the diagnostic parameters between the three methods. 

• We apologize for this statistical error. This has been double-checked by a statistician and the data corrected. 

• Table 6 reveals the association between Plasmodium species and helminthes parasite on malaria parasite density and the nature of anemia. For example, a key observation noted here was that children co-infected with P. falciparum and S. haematobium had similar parasite density and lower hemoglobin level as P. falciparum only infected children.

Comment

Methods:

In the study site description, the authors should add some information on the average rainfall during the season of the collection of samples in mean and specifically for the year

of collection. Using the described malaria slide reading method – what is the obtained limit of detection for the read?

Response: The information has been added (line 151). The limit of detection of the thick blood film slide reading ranges between 50 – 100 parasites/µL (line 239).

Comment

Results: Scholl enrolment is low in the young group – but at what age school is compulsory? Shouldn’t it start at 6 years of age rather than at three years of age?

Response: The basic educational system comprises both the nursery (3 – 5 years) and the primary (≥6 years) levels. Therefore, school enrollment and compulsoriness usually starts at the nursery level grade. However, this has been modified in the text to include the pre-school children (≤5 yr) that were not yet registered in schools at the time of the study.

---

## [Decision Letter · Decision Letter 1]

8 Jun 2023

PONE-D-22-34570R1Epidemiology of malaria, schistosomiasis, and geohelminthiasis amongst children 3-15 years of age during the dry season in Northern CameroonPLOS ONE

Dear Dr. Nongley,

Thank you for submitting your manuscript to PLOS ONE. After careful consideration, we feel that it has merit but does not fully meet PLOS ONE’s publication criteria as it currently stands. Therefore, we invite you to submit a revised version of the manuscript that addresses the points raised during the review process. Please submit your revised manuscript by Jul 23 2023 11:59PM. If you will need more time than this to complete your revisions, please reply to this message or contact the journal office at plosone@plos.org. Please include the following items when submitting your revised manuscript:A rebuttal letter that responds to each point raised by the academic editor and reviewer(s). You should upload this letter as a separate file labeled 'Response to Reviewers'.A marked-up copy of your manuscript that highlights changes made to the original version. You should upload this as a separate file labeled 'Revised Manuscript with Track Changes'.An unmarked version of your revised paper without tracked changes. You should upload this as a separate file labeled 'Manuscript'.If applicable, we recommend that you deposit your laboratory protocols in protocols.io to enhance the reproducibility of your results. Protocols.io assigns your protocol its own identifier (DOI) so that it can be cited independently in the future. For instructions see: https://journals.plos.org/plosone/s/submission-guidelines#loc-laboratory-protocols. Additionally, PLOS ONE offers an option for publishing peer-reviewed Lab Protocol articles, which describe protocols hosted on protocols.io. Read more information on sharing protocols at https://plos.org/protocols?utm_medium=editorial-email&utm_source=authorletters&utm_campaign=protocols.

We look forward to receiving your revised manuscript.

Kind regards,

David Zadock Munisi, Ph.D

Academic Editor

PLOS ONE

Journal Requirements:

Reviewers' comments:

Reviewer's Responses to Questions

**Comments to the Author**

1. If the authors have adequately addressed your comments raised in a previous round of review and you feel that this manuscript is now acceptable for publication, you may indicate that here to bypass the “Comments to the Author” section, enter your conflict of interest statement in the “Confidential to Editor” section, and submit your "Accept" recommendation.

Reviewer #1: All comments have been addressed

Reviewer #2: (No Response)

2. Is the manuscript technically sound, and do the data support the conclusions?

Reviewer #1: Partly

Reviewer #2: Yes

3. Has the statistical analysis been performed appropriately and rigorously? 

Reviewer #1: I Don't Know

Reviewer #2: I Don't Know

4. Have the authors made all data underlying the findings in their manuscript fully available?

Reviewer #1: Yes

Reviewer #2: Yes

5. Is the manuscript presented in an intelligible fashion and written in standard English?

Reviewer #1: Yes

Reviewer #2: Yes

6. Review Comments to the Author

Reviewer #1: The revised version of the manuscript is an improvement in clarity and focus. The authors have largely addressed my concerns.

Reviewer #2: (No Response)

7. PLOS authors have the option to publish the peer review history of their article (what does this mean?). If published, this will include your full peer review and any attached files.

Reviewer #1: No

Reviewer #2: No

---

## [Author Response · Author response to Decision Letter 1]

28 Jun 2023

Response to Reviewers’ Comments

1. “Health education talks on preventive measures for these diseases were delivered before each screening exercise. Furthermore, a questionnaire was administered to each participant.” Don’t you think that by giving them health education pertaining before any screening, it would have affected participants’ responses on the questionnaires? Or what did want to achieve by doing so?

Response: Health talks were delivered as part of explaining the purpose and objectives of the study. This is because participants immediately disperse to their daily activities once their samples have been collected and sometimes most do not even come back for the results or treatment (which is often given to the health worker). Therefore it was logical to deliver the health education lessons once the population assembled for recruitment. Indeed, the fact that the participants responded with poor KAP to malaria and schisto suggests that the health education talks did not impact the response to the questionnaire.

2. I am so worried over the quality of responses from >5 years old children. Do 5 years old children able to comprehend the language in the questionnaire? Have they even ever heard of malaria, schistosomiasis or even remember their deworming history? 

Response: We previously answered this question in the first revision (line 131). Children < 5 years were intermittently assisted to respond to the question by their parents and this was made straightforward because the questionnaire sampling was led and verbally translated by indigenes who speak/understand both English and the local dialect (Fulfude). These children are aware of and have local appellations for these infections as it constitutes their daily health challenges. The deworming history of participants < 5 years was inferred from the parents (where necessary) and further confirmed by the class teachers.

3. In definition of endpoints, have a sentence on symptomatic intestinal schistosomiasis as you did with urogenital schistosomiasis

Response: Thank you for this remark. This has been included (line 304).

4. Line 522, just change “frequent absence of consuming PZQ” to “poor PZQ uptake”

Response: Correction done.

5. Revise comment 9: I have seen the S2 file. But an important piece of information is still missing in your methodology. Simply state the cut-off score of good KAP and bad KAP out of the total 37 variables. And a justification/basis of the decided cut-off

Response: As earlier mentioned (lines 199 – 202), for KAP variables, the cut-off score was between 0 – 1 with 0 = poor KAP and 1 = good KAP, particularly for the Yes/No questions. The basis of the decided cut-off was to categorize the data set and establish relevant epidemiological associations between the level of risk/exposure and outcome.

Conclusion: The authors appreciate the reviewer's comments to improve the scientific quality of the manuscript. We have adequately responded to the comments and hope the manuscript will now be considered for publication.

---

## [Editor Report · Decision Letter 2]

29 Jun 2023

Epidemiology of malaria, schistosomiasis, and geohelminthiasis amongst children 3-15 years of age during the dry season in Northern Cameroon

PONE-D-22-34570R2

Dear Nongley,

We’re pleased to inform you that your manuscript has been judged scientifically suitable for publication and will be formally accepted for publication once it meets all outstanding technical requirements.

Kind regards,

David Zadock Munisi, Ph.D

Academic Editor

PLOS ONE

---

## [Editor Report · Acceptance letter]

21 Jul 2023

PONE-D-22-34570R2 

Epidemiology of malaria, schistosomiasis, and geohelminthiasis amongst children 3-15 years of age during the dry season in Northern Cameroon 

Dear Dr. Nkemngo:

I'm pleased to inform you that your manuscript has been deemed suitable for publication in PLOS ONE. Congratulations! Your manuscript is now with our production department. 

Kind regards, 

on behalf of

Dr. David Zadock Munisi 

Academic Editor

PLOS ONE